# Coarse-grain Fine-grain Coattention Network for Multi-evidence Question Answering

Victor Zhong[1], Caiming Xiong[2], Nitish Shirish Keskar[2], and Richard Socher[2]

[1]Paul G. Allen School of Computer Science & Engineering, University of Washington, Seattle, WA
`vzhong@cs.washington.edu`
[2]Salesforce Research, Palo Alto, CA
`{cxiong, nkeskar, rsocher}@salesforce.com`

## Abstract

End-to-end neural models have made significant progress in question answering, however recent studies show that these models implicitly assume that the answer and evidence appear close together in a single document. In this work, we propose the Coarse-grain Fine-grain Coattention Network (CFC), a new question answering model that combines information from evidence across multiple documents. The CFC consists of a coarse-grain module that interprets documents with respect to the query then finds a relevant answer, and a fine-grain module which scores each candidate answer by comparing its occurrences across all of the documents with the query. We design these modules using hierarchies of coattention and self-attention, which learn to emphasize different parts of the input. On the Qangaroo WikiHop multi-evidence question answering task, the CFC obtains a new state-of-the-art result of 70.6% on the blind test set, outperforming the previous best by 3% accuracy despite not using pretrained contextual encoders.

## 1 Introduction

A requirement of scalable and practical question answering (QA) systems is the ability to reason over multiple documents and combine their information to answer questions. Although existing datasets enabled the development of effective end-to-end neural question answering systems, they tend to focus on reasoning over localized sections of a single document (Hermann et al., 2015; Rajpurkar et al., 2016; 2018; Trischler et al., 2017). For example, Min et al. (2018) find that 90% of the questions in the Stanford Question Answering Dataset are answerable given 1 sentence in a document. In this work, we instead focus on multi-evidence QA, in which answering the question requires aggregating evidence from multiple documents (Welbl et al., 2018; Joshi et al., 2017).

Our multi-evidence QA model, the Coarse-grain Fine-grain Coattention Network (CFC), selects among a set of candidate answers given a set of support documents and a query. The CFC is inspired by coarse-grain reasoning and fine-grain reasoning. In coarse-grain reasoning, the model builds a coarse summary of support documents conditioned on the query without knowing what candidates are available, then scores each candidate. In fine-grain reasoning, the model matches specific fine-grain contexts in which the candidate is mentioned with the query in order to gauge the relevance of the candidate. These two strategies of reasoning are respectively modeled by the coarse-grain and fine-grain modules of the CFC. Each module employs a novel hierarchical attention — a hierarchy of coattention and self-attention — to combine information from the support documents conditioned on the query and candidates. Figure 1 illustrates the architecture of the CFC.

The CFC achieves a new state-of-the-art result on the blind Qangaroo WikiHop test set of 70.6% accuracy, beating previous best by 3% accuracy despite not using pretrained contextual encoders. In addition, on the TriviaQA multi-paragraph question answering task (Joshi et al., 2017), reranking

---

Most of this work was done while Victor Zhong was at Salesforce Research.

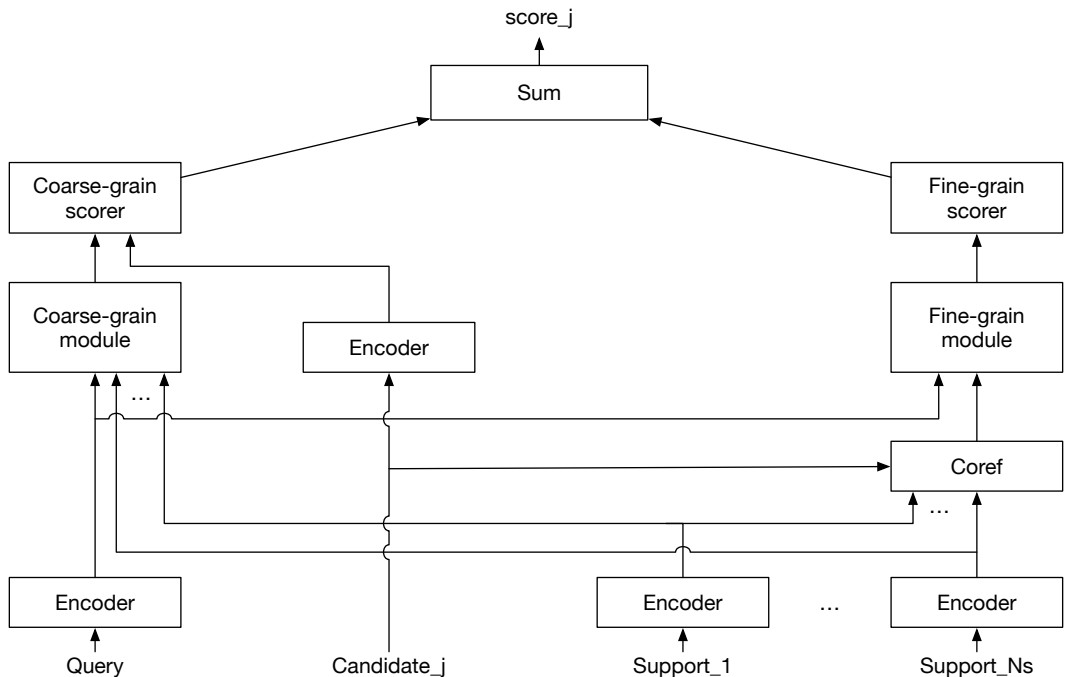

Figure 1: The Coarse-grain Fine-grain Coattention Network.

outputs from a traditional span extraction model (Clark & Gardner, 2018) using the CFC improves exact match accuracy by 3.1% and F1 by 3.0%.

Our analysis shows that components in the attention hierarchies of the coarse and fine-grain modules learn to focus on distinct parts of the input. This enables the CFC to more effectively represent a large collection of long documents. Finally, we outline common types of errors produced by CFC, caused by difficulty in aggregating large quantity of references, noise in distant supervision, and difficult relation types.

## 2 COARSE-GRAIN FINE-GRAIN COATTENTION NETWORK

The coarse-grain module and fine-grain module of the CFC correspond to coarse-grain reasoning and fine-grain reasoning strategies. The coarse-grain module summarizes support documents without knowing the candidates: it builds codependent representations of support documents and the query using coattention, then produces a coarse-grain summary using self-attention. In contrast, the fine-grain module retrieves specific contexts in which each candidate occurs: it identifies coreferent mentions of the candidate, then uses coattention to build codependent representations between these mentions and the query. While low-level encodings of the inputs are shared between modules, we show that this division of labour allows the attention hierarchies in each module to focus on different parts of the input. This enables the model to more effectively represent a large number of potentially long support documents.

Suppose we are given a query, a set of $N_s$ support documents, and a set of $N_c$ candidates. Without loss of generality, let us consider the $i$th document and the $j$th candidate. Let $L_q \in \mathbb{R}^{T_q \times d_{\mathrm{emb}}}$, $L_s \in \mathbb{R}^{T_s \times d_{\mathrm{emb}}}$, and $L_c \in \mathbb{R}^{T_c \times d_{\mathrm{emb}}}$ respectively denote the word embeddings of the query, the $i$th support document, and the $j$th candidate answer. Here, $T_q$, $T_s$, and $T_c$ are the number of words in the corresponding sequence. $d_{\mathrm{emb}}$ is the size of the word embedding. We begin by encoding each sequence using a bidirectional Gated Recurrent Units (GRUs) (Cho et al., 2014).

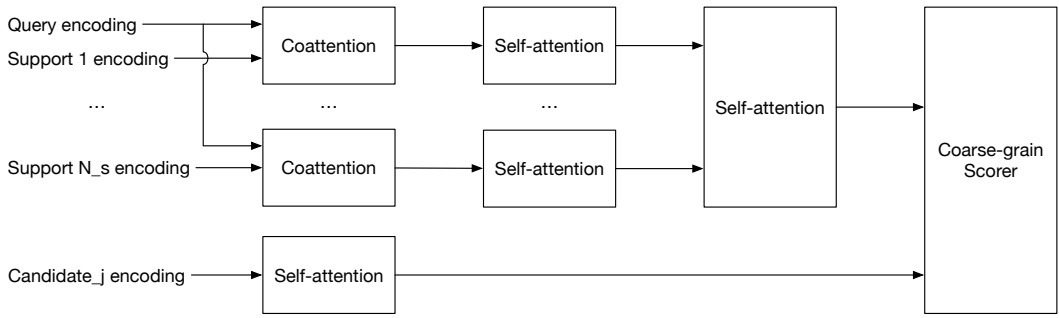

Figure 2: Coarse-grain module.

$$E_q = \mathrm{BiGRU}\left(\tanh(W_q L_q + b_q)\right) \in \mathbb{R}^{T_q \times d_{\mathrm{hid}}} \tag{1}$$

$$E_s = \mathrm{BiGRU}\left(L_s\right) \in \mathbb{R}^{T_s \times d_{\mathrm{hid}}} \tag{2}$$

$$E_c = \mathrm{BiGRU}\left(L_c\right) \in \mathbb{R}^{T_c \times d_{\mathrm{hid}}} \tag{3}$$

Here, $E_q$, $E_s$, and $E_c$ are the encodings of the query, support, and candidate. $W_q$ and $b_q$ are parameters of a query projection layer. $d_{\mathrm{hid}}$ is the size of the bidirectional GRU.

## 2.1 COARSE-GRAIN MODULE

The coarse-grain module of the CFC, shown in Figure 2, builds codependent representations of support documents $E_s$ and the query $E_q$ using coattention, and then summarizes the coattention context using self-attention to compare it to the candidate $E_c$. Coattention and similar techniques are crucial to single-document question answering models (Xiong et al., 2017; Wang & Jiang, 2017; Seo et al., 2017). We start by computing the affinity matrix between the document and the query as

$$A = E_s(E_q)^{\mathsf{T}} \in \mathbb{R}^{T_s \times T_q} \tag{4}$$

The support summary vectors and query summary vectors are defined as

$$S_s = \mathrm{softmax}\left(A\right) E_q \in \mathbb{R}^{T_s \times d_{\mathrm{hid}}} \tag{5}$$

$$S_q = \mathrm{softmax}\left(A^{\mathsf{T}}\right) E_s \in \mathbb{R}^{T_q \times d_{\mathrm{hid}}} \tag{6}$$

where $\mathrm{softmax}(X)$ normalizes $X$ column-wise. We obtain the document context as

$$C_s = \mathrm{BiGRU}\left(S_q \, \mathrm{softmax}\left(A\right)\right) \in \mathbb{R}^{T_s \times d_{\mathrm{hid}}} \tag{7}$$

The coattention context is then the feature-wise concatenation of the document context $C_s$ and the document summary vector $S_s$.

$$U_s = [C_s; S_s] \in \mathbb{R}^{T_s \times 2d_{\mathrm{hid}}} \tag{8}$$

For ease of exposition, we abbreviate coattention, which takes as input a document encoding $E_s$ and a query encoding $E_q$ and produces the coattention context $U_s$, as

$$\mathrm{Coattn}\left(E_s, E_q\right) \to U_s \tag{9}$$

Next, we summarize the coattention context — a codependent encoding of the supporting document and the query — using hierarchical self-attention. First, we use self-attention to create a fixed-length summary vector of the coattention context. We compute a score for each position of the

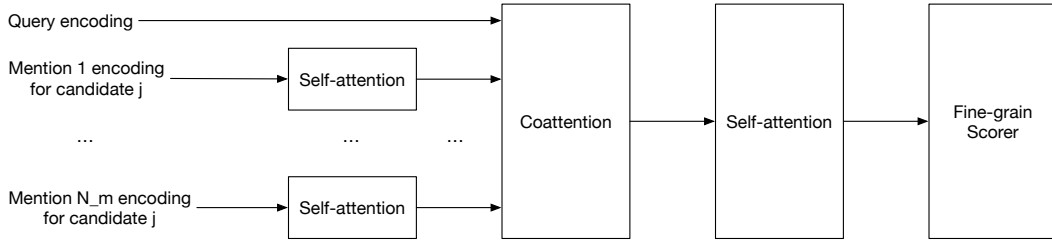

Figure 3: The fine-grain module of the CFC.

coattention context using a two-layer multi-layer perceptron (MLP). This score is normalized and used to compute a weighted sum over the coattention context.

$$
\begin{align}
a_{si} &= \tanh\left(W_2 \tanh\left(W_1 U_{si} + b_1\right) + b_2\right) \in \mathbb{R} \tag{10}\\
\hat{a}_s &= \mathrm{softmax}(a_s) \tag{11}\\
G_s &= \sum_i^{T_s} \hat{a}_{si} U_{si} \in \mathbb{R}^{2d_{\mathrm{hid}}} \tag{12}
\end{align}
$$

Here, $a_{si}$ and $\hat{a}_{si}$ are respectively the unnormalized and normalized score for the $i$th position of the coattention context. $W_2$, $b_2$, $W_1$, and $b_1$ are parameters for the MLP scorer. $U_{si}$ is the $i$th position of the coattention context. We abbreviate self-attention, which takes as input a sequence $U_s$ and produces the summary conditioned on the query $G_s$, as

$$
\mathrm{Selfattn}\left(U_s\right) \rightarrow G_s \tag{13}
$$

Recall that $G_s$ provides the summary of the $i$th of $N_s$ support documents. We apply another self-attention layer to compute a fixed-length summary vector of all support documents. This summary is then multiplied with the summary of the candidate answer to produce the coarse-grain score. Let $G \in \mathbb{R}^{N_s \times 2d_{\mathrm{hid}}}$ represent the sequence of summaries for all support documents. We have

$$
\begin{align}
G_c &= \mathrm{Selfattn}\left(E_c\right) \in \mathbb{R}^{d_{\mathrm{hid}}} \tag{14}\\
G' &= \mathrm{Selfattn}\left(G\right) \in \mathbb{R}^{2d_{\mathrm{hid}}} \tag{15}\\
y_{\mathrm{coarse}} &= \tanh\left(W_{\mathrm{coarse}} G' + b_{\mathrm{coarse}}\right) G_c \in \mathbb{R} \tag{16}
\end{align}
$$

where $E_c$ and $G_c$ are respectively the encoding and the self-attention summary of the candidate. $G'$ is the fixed-length summary vector of all support documents. $W_{\mathrm{coarse}}$ and $b_{\mathrm{coarse}}$ are parameters of a projection layer that reduces the support documents summary from $\mathbb{R}^{2d_{\mathrm{hid}}}$ to $\mathbb{R}^{d_{\mathrm{hid}}}$.

## 2.2 CANDIDATE-DEPENDENT FINE-GRAIN MODULE

In contrast to the coarse-grain module, the fine-grain module, shown in Figure 3, finds the specific context in which the candidate occurs in the supporting documents using coreference resolution [1]. Each mention is then summarized using a self-attention layer to form a mention representation. We then compute the coattention between the mention representations and the query. This coattention context, which is a codependent encoding of the mentions and the query, is again summarized via self-attention to produce a fine-grain summary to score the candidate.

Let us assume that there are $m$ mentions of the candidate in the $i$th support document. Let the $k$th mention corresponds to the $i_{\mathrm{start}}$ to $i_{\mathrm{end}}$ tokens in the support document. We represent this mention using self-attention over the span of the support document encoding that corresponds to the mention.

---

[1] We use simple lexical matching between candidates and support documents. Details are found in A.1 of the Appendix.

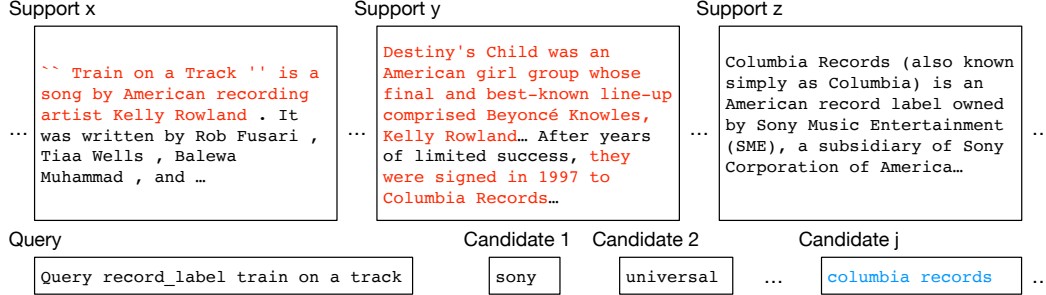

Figure 4: An example from the Qangaroo WikiHop QA task. The relevant multiple pieces of evidence required to answer the question is shown in red. The correct answer is shown in blue.

$$M_k = \text{Selfattn}\left(E_s[i_{\text{start}} : i_{\text{end}}]\right) \in \mathbb{R}^{d_{\text{hid}}} \tag{17}$$

Suppose that there are $N_m$ mentions of the candidate in total. We extract each mention representation using self-attention to produce a sequence of mention representations $M \in \mathbb{R}^{N_m \times d_{\text{hid}}}$. The coattention context and summary of these mentions $M$ with respect to the query $E_q$ are

$$U_m = \text{Coattn}\left(M, E_q\right) \in \mathbb{R}^{N_m \times 2d_{\text{hid}}} \tag{18}$$

$$G_m = \text{Selfattn}\left(U_m\right) \in \mathbb{R}^{2d_{\text{hid}}} \tag{19}$$

We use a linear layer to determine the fine-grain score of the candidate

$$y_{\text{fine}} = W_{\text{fine}} G_m + b_{\text{fine}} \in \mathbb{R} \tag{20}$$

## 2.3 SCORE AGGREGATION

We take the sum of the coarse-grain score and the fine-grain score, $y = y_{\text{coarse}} + y_{\text{fine}}$, as the score for the candidate. Recall that our earlier presentation is with respect to the $j$th out of $N_c$ candidates. We combine each candidate score to form the final score vector $Y \in \mathbb{R}^{N_c}$. The model is trained using cross-entropy loss.

## 3 EXPERIMENTS

We evaluate the CFC on two tasks to evaluate its effectiveness. The first task is multi-evidence question answering on the unmasked and masked version of the WikiHop dataset (Welbl et al., 2018). The second task is the multi-paragraph extractive question answering task TriviaQA, which we frame as a span reranking task (Joshi et al., 2017). On the former, the CFC achieves a new state-of-the-art result. On the latter, reranking the outputs of a span-extraction model (Clark & Gardner, 2018) using the CFC results in significant performance improvement.

## 3.1 MULTI-EVIDENCE QUESTION ANSWERING ON WIKIHOP

Welbl et al. (2018) proposed the Qangaroo WikiHop task to facilitate the study of multi-evidence question answering. This dataset is constructed by linking entities in a document corpus (Wikipedia) with a knowledge base (Wikidata). This produces a bipartite graph of documents and entities, an edge in which marks the occurrence of an entity in a document. A knowledge base fact triplet consequently corresponds to a path from the subject to the object in the resulting graph. The documents along this path compose the support documents for the fact triplet. The Qangaroo WikiHop task, shown in Figure 4, is as follows: given a query, that is, the subject and relation of a fact triplet, a set

| Model | Masked Dev | Dev | Test |
|---|---|---|---|
| CFC (ours) | **72.1%** | **66.4%** | **70.6%** |
| Enitity-GCN (Cao et al., 2018) | 70.5% | 64.8% | 67.6% |
| MHQA-GRN (Song et al., 2018) | | 62.8% | 65.4% |
| Jenga (Facebook AI Research*, 2018) | | | 65.3% |
| Vanilla Coattention Model (NTU*, 2018) | | | 59.9% |
| Coref GRU (Dhingra et al., 2018) | | 56.0% | 59.3% |
| BiDAF Baseline (Welbl et al., 2018) | 54.5% | | 42.9% |

Table 1: Model accuracy on the WikiHop leaderboard at the time of submission on September 14, 2018. Missing entries indicate that the published entry did not include the corresponding score. * indicates that the work has not been published.

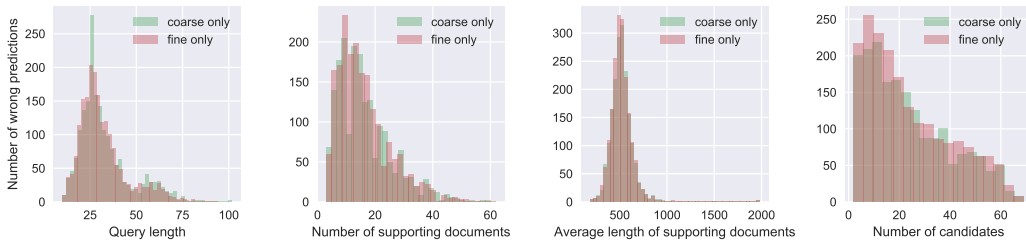

Figure 5: WikiHop dev errors across query lengths, support documents lengths, number of support documents, and number of candidates for the coarse-grain-only and fine-grain-only models.

of plausible candidate objects, and the corresponding support documents for the candidates, select the correct candidate as the answer.

The unmasked version of WikiHop represents candidate answers with original text while the masked version replaces them with randomly sampled placeholders in order to remove correlation between frequent answers and support documents. Official blind, held-out test evaluation is performed using the unmasked version. We tokenize the data using Stanford CoreNLP (Manning et al., 2014). We use fixed GloVe embeddings (Pennington et al., 2014) as well as character ngram embeddings (Hashimoto et al., 2017). We split symbolic query relations into words. All models are trained using ADAM (Kingma & Ba, 2015). We list detailed experiment setup and hyperparemeters of the best-performing model in A.2 of the Appendix.

We compare the performance of the CFC to other models on the WikiHop leaderboard in Table 1. The CFC achieves state-of-the-art results on both the masked and unmasked versions of WikiHop. In particular, on the blind, held-out WikiHop test set, the CFC achieves a new best accuracy of 70.6%. The previous state-of-the-art result by Cao et al. (2018) uses pretrained contextual encoders, which has led to consistent improvements across NLP tasks (Peters et al., 2018). We outperform this result by 3% despite not using pretrained contextual encoders [2]. In addition, we show that the division of labour between the coarse-grain module and the fine-grain module allows the attention hierarchies of each module to focus on different parts of the input. This enables the CFC to more effectively model the large collection of potentially long documents found in WikiHop.

## 3.2 RERANKING EXTRACTIVE QUESTION ANSWERING ON TRIVIAQA

To further study the effectiveness of our model, we also experiment on TriviaQA (Joshi et al., 2017), another large-scale question answering dataset that requires aggregating evidence from multiple sentences. Similar to Hu et al. (2018b); Wang et al. (2018), we decompose the original TriviaQA task into two subtasks: proposing plausible candidate answers and reranking candidate answers.

---

[2] Adding contextual encoders is computationally challenging as the CFC models the entire document context unlike (Cao et al., 2018). It is an area for future work.

| Answerable | % of data | Before reranking | | After reranking | |
|---|---|---|---|---|---|
| | | EM | F1 | EM | F1 |
| Answerable | 86.8% | 59.8% | 64.5% | 63.2% | 67.8% |
| Unanswerable | 13.2% | 17.5% | 22.2% | 17.9% | 22.9% |
| Total | 100% | 54.0% | 58.7% | 57.1% | 61.7% |

Table 2: Answer reranking results on the dev split of TriviaQA Wikipedia. We use the BiDAF++ model with the merge method of span scoring by Clark & Gardner (2018) to propose candidate answers, which are subsequently reranked using the CFC. "Answerable" indicates that the candidate answers proposed contains at least one correct answer. "Unanswerable" indicates that none of the candidate answers proposed are correct.

We address the first subtask using BiDAF++, a competitive span extraction question answering model by Clark & Gardner (2018) and the second subtask using the CFC. To compute the candidate list for reranking, we obtain the top 50 answer candidates from BiDAF++. During training, we use the answer candidate that gives the maximum F1 as the gold label for training the CFC.

Our experimental results in Table 2 show that reranking using the CFC provides consistent performance gains over only using the span extraction question answering model. In particular, reranking using the CFC improves performance regardless of whether the candidate answer set obtained from the span extraction model contains correct answers. On the whole TriviaQA dev set, reranking using the CFC results in a gain of 3.1% EM and 3.0% F1, which suggests that the CFC can be used to further refine the outputs produced by span extraction question answering models.

| Model | Dev | Δ Dev |
|---|---|---|
| CFC | 66.4% | |
| -coarse | 61.9% | -4.5% |
| -fine | 63.6% | -2.8% |
| -selfattn | 64.8% | -1.6% |
| -bidir | 65.4% | -1.0% |
| -encoder | 61.3% | -5.1% |

Table 3: Ablation study on the WikiHop dev set. The rows respectively correspond to the removal of coarse-grain module, the removal of fine-grain module, the replacement of self-attention with average pooling, the replacement of bidir. with unidir. GRUs, and the replacement of encoder GRUs with projection over word embeddings.

### 3.3 ABLATION STUDY

Table 3 shows the performance contributions of the coarse-grain module, the fine-grain module, as well as model decisions such as self-attention and bidirectional GRUs. Both the coarse-grain module and the fine-grain module significantly contribute to model performance. Replacing self-attention layers with mean-pooling and the bidirectional GRUs with unidirectional GRUs result in less performance degradation. Replacing the encoder with a projection over word embeddings result in significant performance drop, which suggests that contextual encodings that capture positional information is crucial to this task.

Figure 5 shows the distribution of model prediction errors across various lengths of the dataset for the coarse-grain-only model (-fine) and the fine-grain-only model (-coarse). The fine-grain-only model under-performs the coarse-grain-only model consistently across almost all length measures. This is likely due to the difficulty of coreference resolution of candidates in the support documents — the technique we use of exact lexical matching tends to produce high precision and low recall. However, the fine-grain-only model matches or outperforms the coarse-grain-only model on examples with a large number of support documents or with long support documents. This is likely because the entity-matching coreference resolution we employ captures intra-document and inter-document dependencies more precisely than hierarchical attention.

### 3.4 QUALITATIVE ANALYSIS

We examine the hierarchical attention maps produced by the CFC on examples from the Wiki-Hop development set. We find that coattention layers consistently focus on phrases that are similar between the document and the query, while lower level self-attention layers capture phrases that characterize the entity described by the document. Because these attention maps are very large, we do not include them in the main text and instead refer readers to A.3 of the Appendix.

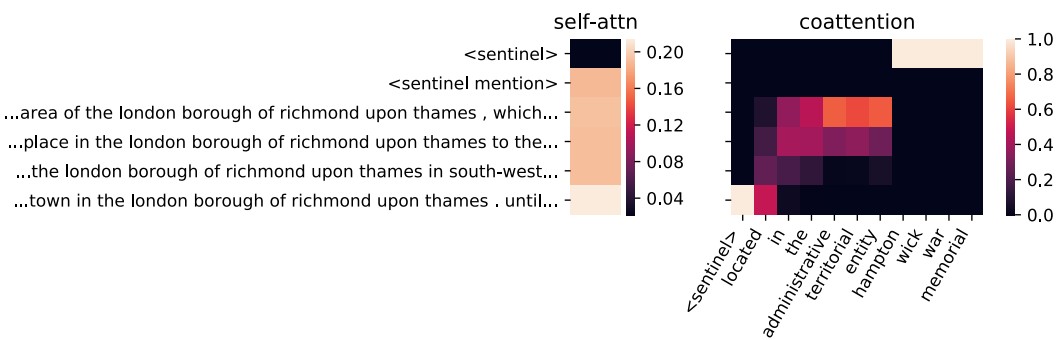

Figure 7: Fine-grain coattention and self-attention scores for for the query `located_in_the_administrative_territorial_entity hampton wick war memorial`, for which the answer is "London borough of Richmond Upon Thames". The coattention tends to align the relation part of the query to the context in which the mention occurs in the text. The first, second, and fourth mentions respectively describe Hampton Wicks, Hampton Hills, and Teddington — all of which are located in Richmond upon Thames. The third describes Richmond upon Thames itself.

Coarse-grain summary self-attention, described in equation 15, tends to focus on support documents that present information relevant to the object in the query. Figure 6 illustrates an example of this in which the self-attention focuses on documents relevant to the literary work "The Troll", namely those about The Troll, its author Julia Donaldson, and Old Norse.

In contrast, fine-grain coattention over mention representations, described in equation 19, tends to focus on the relation part of the query. Figure 7 illustrates an example of this in which the coattention focuses on the relationship between the mentions and the phrase "located in the administrative territorial entity". Attention maps of more examples can be found in A.3 of the Appendix.

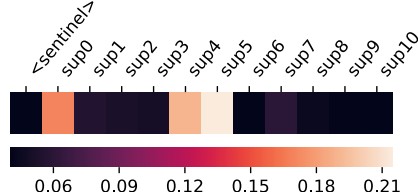

Figure 6: Coarse-grain summary self-attention scores for the query `country_of_origin the troll`, for which the answer is "United Kingdom". The summary self-attention tends to focus on documents relevant to the subject in the query. The top three support documents 2, 4, 5 respectively present information about the literary work The Troll, its author Julia Donaldson, and Old Norse.

## 3.5 ERROR ANALYSIS

We examine 100 errors the CFC produced on the WikiHop development set and categorize them into four types. We list identifiers and examples of these errors in A.4 of the Appendix. The first type (42% of errors) results from the model aggregating the wrong reference. For example, for the query `country_of_citizenship jamie burnett`, the model correctly attends to the documents about Jamie Burnett being born in South Larnarkshire and about Lanarkshire being in Scotland. However it wrongly focuses on the word "england" in the latter document instead of the answer "scotland". We hypothesize that ways to reduce this type of error include using more robust pretrained contextual encoders (McCann et al., 2017; Peters et al., 2018) and coreference resolution. The second type (28% of errors) results from questions that are not answerable. For example, the support documents do not provide the narrative location of the play "The Beloved Vagabond" for the query `narrative_location the beloved vagabond`. The third type (22% of errors) results from queries that yield multiple correct answers. An example is the query `instance_of qilakitsoq`, for which the model predicts "archaeological site", which is more specific than the answer "town". The second and third types of errors underscore the difficulty of using distant supervision to create large-scale datasets such as WikiHop. The fourth type (8% of errors) results from complex relation types such as `parent_taxon` which are difficult to interpret using pretrained word embeddings. One method to alleviate this type of errors is to embed relations using tunable symbolic embeddings as well as fixed word embeddings.

## 4 RELATED WORK

**Question answering and information aggregation tasks.** QA tasks span a variety of sources such as Wikipedia (Yang et al., 2015; Rajpurkar et al., 2016; 2018; Hewlett et al., 2016; Joshi et al., 2017; Welbl et al., 2018), news articles (Hermann et al., 2015; Trischler et al., 2017), books (Richardson et al., 2013), and trivia (Iyyer et al., 2014). Most QA tasks seldom require reasoning over multiple pieces of evidence. In the event that such reasoning is required, it typically arises in the form of coreference resolution within a single document (Min et al., 2018). In contrast, the Qangaroo WikiHop dataset encourages reasoning over multiple pieces of evidence across documents due to its construction. A similar task that also requires aggregating information from multiple documents is query-focused multi-document summarization, in which a model summarizes a collection of documents given an input query (Dang, 2006; Gupta et al., 2007; Lu et al., 2013).

**Question answering models.** The recent development of large-scale QA datasets has led to a host of end-to-end QA models. These include early document attention models for cloze-form QA (Chen et al., 2015), multi-hop memory networks (Weston et al., 2015; Sukhbaatar et al., 2015; Kumar et al., 2016), as well as cross-sequence attention models for span-extraction QA. Variations of cross-sequence attention include match-LSTM (Wang & Jiang, 2017), coattention (Xiong et al., 2017; 2018), bidirectional attention (Seo et al., 2017), and query-context attention (Yu et al., 2018). Recent advances include the use of reinforcement learning to encourage the exploration of close answers that may have imprecise span match (Xiong et al., 2018; Hu et al., 2018a), the use of convolutions and self-attention to model local and global interactions (Yu et al., 2018), as well as the addition of reranking models to refine span-extraction output (Wang et al., 2018; Hu et al., 2018b). Our work builds upon prior work on single-document QA and generalizes to multi-evidence QA across documents.

**Attention as information aggregation.** Neural attention has been successfully applied to a variety of tasks to summarize and aggregate information. Bahdanau et al. (2015) demonstrate the use of attention over the encoder to capture soft alignments for machine translation. Similar types of attention has also been used in relation extraction (Zhang et al., 2017), summarization (Rush et al., 2015), and semantic parsing (Dong & Lapata, 2018). Coattention as a means to encode codependent representations between two inputs has also been successfully applied to visual question answering (Lu et al., 2016) in addition to textual question answering. Self-attention has similarly been shown to be effective as a means to combine information in textual entailment (Shen et al., 2018; Deunsol Yoon, 2018), coreference resolution (Lee et al., 2017), dialogue state-tracking (Zhong et al., 2018), machine translation (Vaswani et al., 2017), and semantic parsing (Kitaev & Klein, 2018). In the CFC, we present a novel way to combine self-attention and coattention in a hierarchy to build effective conditional and codependent representations of a large number of potentially long documents.

**Coarse-to-fine modeling.** Hierarchical coarse-to-fine modeling, which gradually introduces complexity, is an effective technique to model long documents. Petrov (2009) provides a detailed overview of this technique and demonstrates its effectiveness on parsing, speech recognition, and machine translation. Neural coarse-to-fine modeling has also been applied to question answering (Choi et al., 2017; Min et al., 2018; Swayamdipta et al., 2018) and semantic parsing (Dong & Lapata, 2018). The coarse and fine-grain modules of the CFC similarly focus on extracting coarse and fine representations of the input. Unlike previous work in which a coarse module precedes a fine module, the modules in the CFC are complementary.

## 5 CONCLUSION

We presented CFC, a new state-of-the-art model for multi-evidence question answering inspired by coarse-grain reasoning and fine-grain reasoning. On the WikiHop question answering task, the CFC achieves 70.6% test accuracy, outperforming previous methods by 3% accuracy. We showed in our analysis that the complementary coarse-grain and fine-grain modules of the CFC focus on different aspects of the input, and are an effective means to represent large collections of long documents.

ACKNOWLEDGEMENT

The authors thank Luke Zettlemoyer for his feedback and advice and Sewon Min for her help in preprocessing the TriviaQA dataset.

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

# A    APPENDIX

## A.1    COREFERENCE RESOLUTION

In this work, we use simple lexical matching instead of using full-scale coreference resolution systems. The integration of the latter remains a direction for future work. To perform simple lexical matching for a given candidate, we first tokenize the document as well as the candidate. Each time the candidate tokens occur consequetively in the document, we extract the corresponding token span as a coreference mention.

## A.2    EXPERIMENT SETUP

For the best-performing model, we train the  CFC using Adam (Kingma & Ba, 2015) for a maximum of 50 epochs with a batch size of 80 examples. We use an initial learning rate of $10^{-3}$ with $(\beta_1, \beta_2) = (0.9, 0.999)$ and employ a cosine learning rate decay Loshchilov & Hutter (2017) over the maximum budget. We find this approach to outperform a development set-based annealing heuristic as well as those based on piecewise-constant approximations. We evaluate the accuracy of the model on the development set every epoch, and evaluate the model that obtained the best accuracy on the development set on the held-out test set. We present the convergence plot in Figure 8.

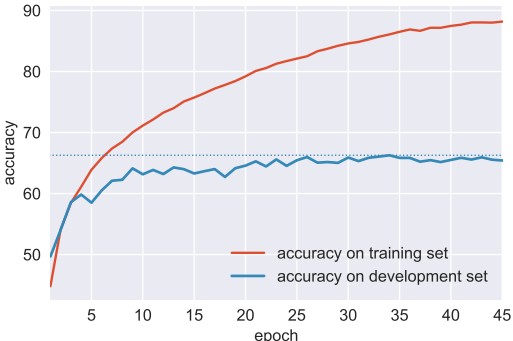

Figure 8: Accuracy convergence plot.

We use a embedding size of $d_{\text{emb}} = 400$, 300 of which are from GloVe vectors (Pennington et al., 2014) and 100 of which are from character ngram vectors (Hashimoto et al., 2017). The embeddings are fixed and not tuned during training. All GRUs have a hidden size of $d_{\text{hid}} = 100$. We regularize the model using dropout (Srivastava et al., 2014) at several locations in the model: after the embedding layer with a rate of 0.3, encoders with a rate of 0.3, coattention layers with a rate of 0.2, and self-attention layers with a rate of 0.2. We also apply word dropout with a rate of 0.25 (Zhang et al., 2017; Zhong et al., 2018). The values for the dropout rates are coarsely tuned and we find that performance is more sensitive to word dropout than other dropout.

## A.3    ATTENTION MAPS

This section includes attention maps produced by the CFC on the development split of WikiHop. We include the fine-grain mention self-attention and coattention, the coarse-grain summary self-attention, and the document self-attention and coattention for the top scoring supporing documents, ranked by the summary self-attention score. The query can be found in the coattention maps. We use the answer as the title of the subsection.

### A.3.1 HOUSE OF VALOIS

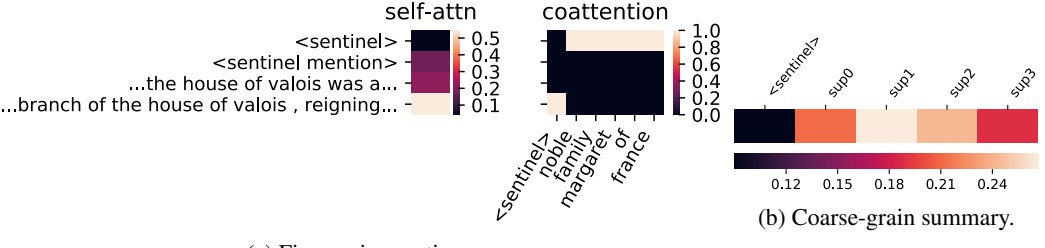

(a) Fine-grain mentions.

(b) Coarse-grain summary.

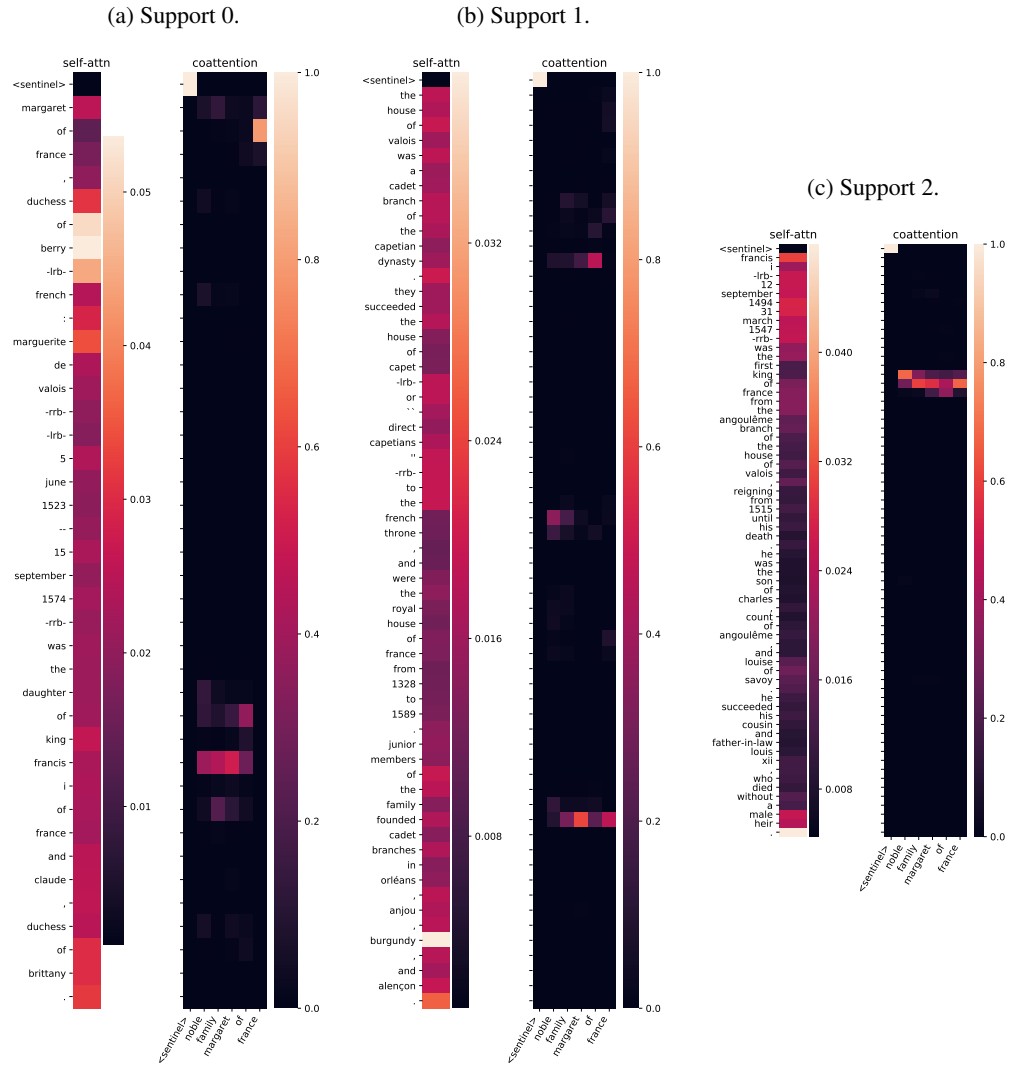

(a) Support 0.

(b) Support 1.

(c) Support 2.

Figure 10: Top supporting documents.

### A.3.2 GERMAN EMPIRE

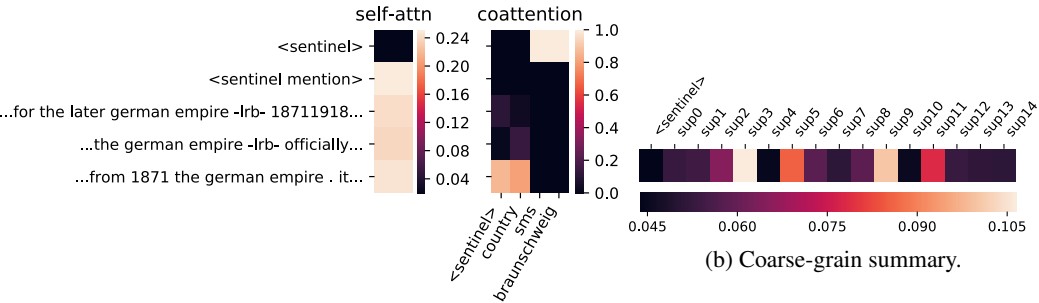

(a) Fine-grain mentions.

(b) Coarse-grain summary.

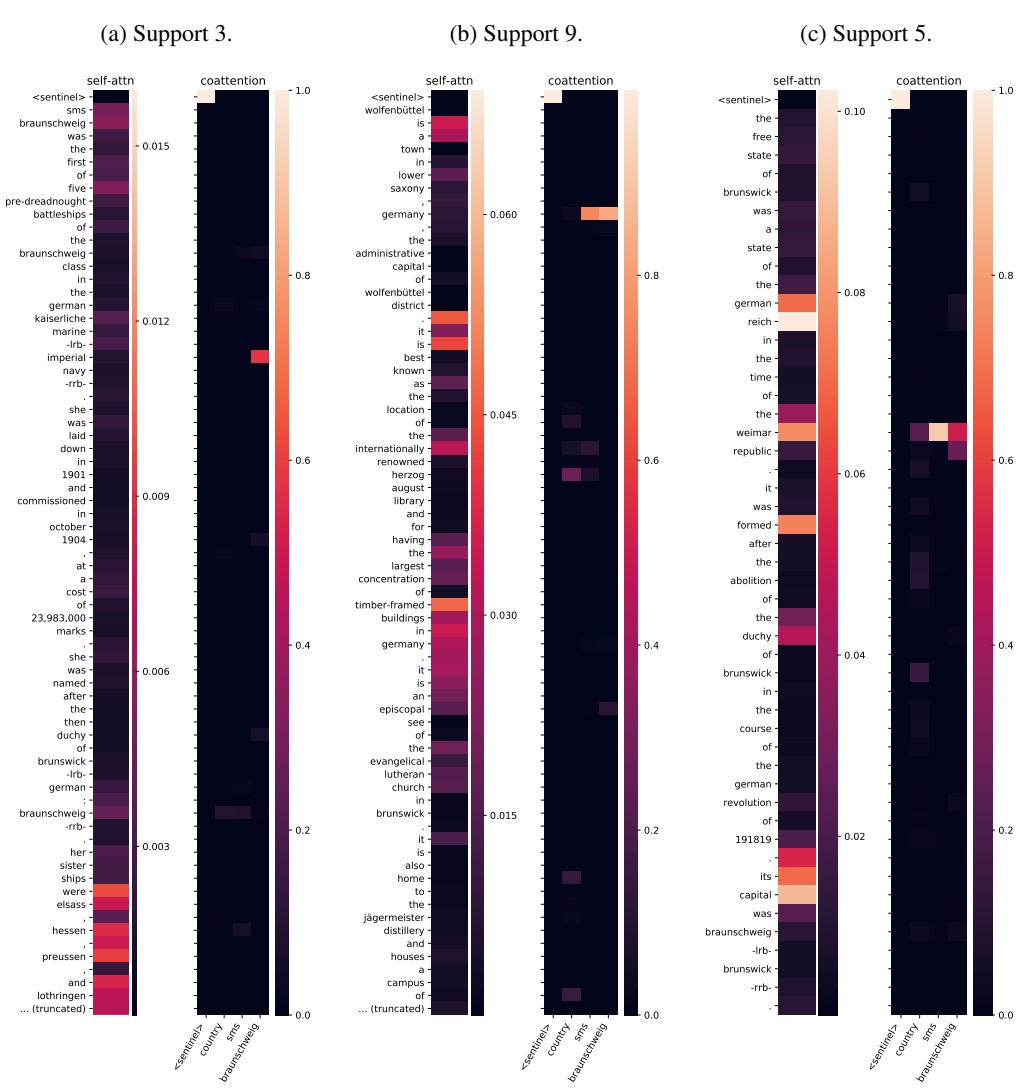

(a) Support 3.    (b) Support 9.    (c) Support 5.

Figure 12: Top supporting documents.

### A.3.3 UNITED KINGDOM

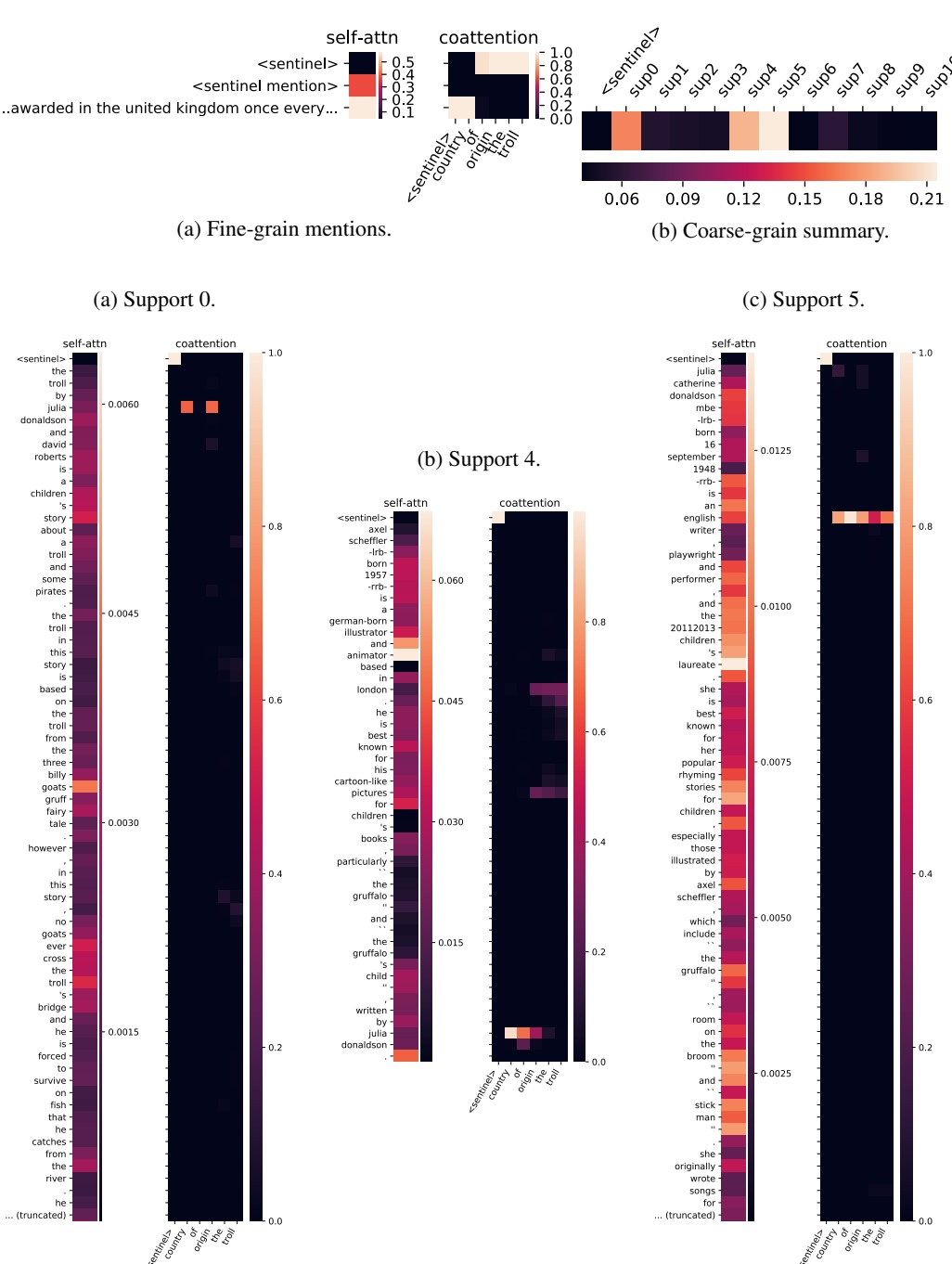

(a) Fine-grain mentions.

(b) Coarse-grain summary.

(a) Support 0.

(c) Support 5.

(b) Support 4.

Figure 14: Top supporting documents.

### A.3.4 LONDON BOROUGH OF RICHMOND UPON THAMES

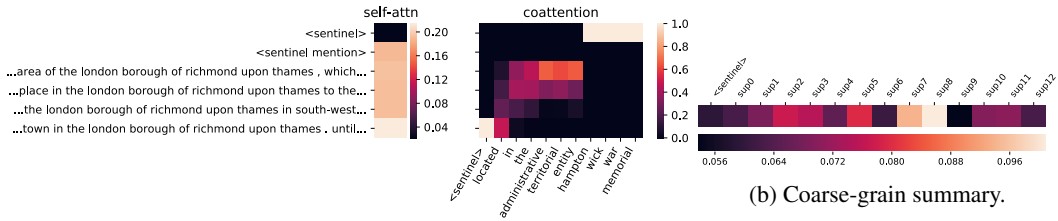

(a) Fine-grain mentions.

(b) Coarse-grain summary.

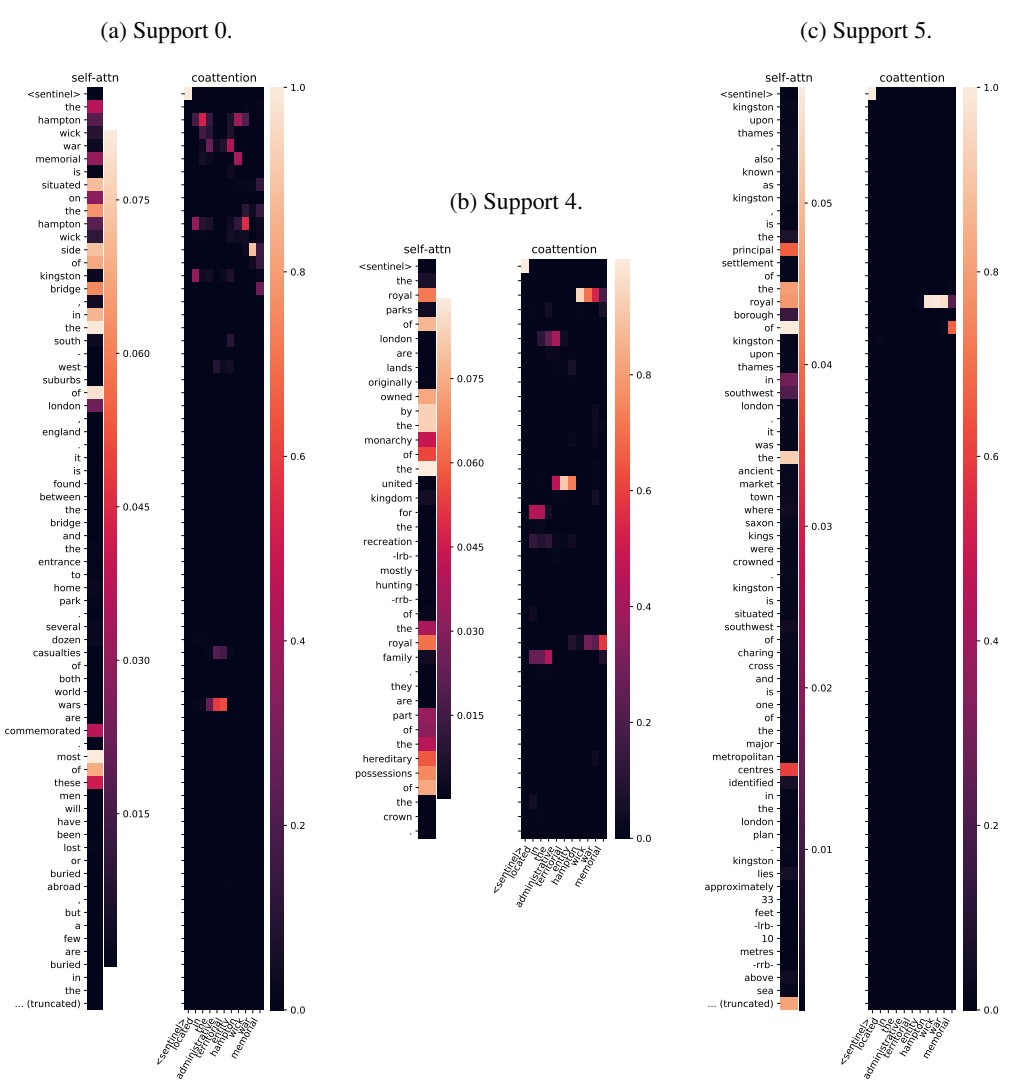

(a) Support 0.

(b) Support 4.

(c) Support 5.

Figure 16: Top supporting documents.

## A.4 ERROR ANALYSIS

This section includes identifiers and examples of the unanswerable questions we found in the development set during error analysis. In particular, these corresponds to 100 randomly sampled errors made by the CFC on the dev split of WikiHop.

**Type 1 Error** WH_dev_1, WH_dev_5, WH_dev_8, WH_dev_29, WH_dev_30, WH_dev_36, WH_dev_40, WH_dev_66, WH_dev_71, WH_dev_76, WH_dev_77, WH_dev_78, WH_dev_80, WH_dev_95, WH_dev_96, WH_dev_97, WH_dev_107, WH_dev_108, WH_dev_109, WH_dev_111, WH_dev_113, WH_dev_114, WH_dev_116, WH_dev_125, WH_dev_143, WH_dev_148, WH_dev_151, WH_dev_156, WH_dev_161, WH_dev_162, WH_dev_164, WH_dev_175, WH_dev_188, WH_dev_190, WH_dev_191, WH_dev_193, WH_dev_196, WH_dev_198, WH_dev_212, WH_dev_215, WH_dev_224, WH_dev_256

**Type 2 Error** WH_dev_35, WH_dev_68, WH_dev_69, WH_dev_81, WH_dev_87, WH_dev_89, WH_dev_98, WH_dev_123, WH_dev_139, WH_dev_150, WH_dev_153, WH_dev_154, WH_dev_155, WH_dev_158, WH_dev_160, WH_dev_168, WH_dev_200, WH_dev_203, WH_dev_205, WH_dev_208, WH_dev_218, WH_dev_221, WH_dev_226, WH_dev_228, WH_dev_230, WH_dev_239, WH_dev_252, WH_dev_260

**Type 3 Error** WH_dev_13, WH_dev_16, WH_dev_18, WH_dev_23, WH_dev_32, WH_dev_58, WH_dev_65, WH_dev_83, WH_dev_86, WH_dev_100, WH_dev_107, WH_dev_140, WH_dev_144, WH_dev_172, WH_dev_176, WH_dev_186, WH_dev_189, WH_dev_220, WH_dev_222, WH_dev_233, WH_dev_243, WH_dev_262

**Type 4 Error** WH_dev_14, WH_dev_47, WH_dev_115, WH_dev_120, WH_dev_133, WH_dev_134, WH_dev_142, WH_dev_234

A.4.1 TYPE 1 ERROR: AGGREGATION OF WRONG REFERENCE

**Total** $\frac{42}{100}$

**Query** `country_of_citizenship jamie burnett`

**Candidates** british empire, england, london, scotland, united kingdom

**Answer** scotland

**Prediction** england

**Support documents** Jamie Burnett ( born 16 September 1975 ) is a professional snooker player from Hamilton , South Lanarkshire .

Glasgow is the largest city in Scotland, and third largest in the United Kingdom. Historically part of Lanarkshire, it is now one of the 32 council areas of Scotland. It is situated on the River Clyde in the countrys West Central Lowlands. Inhabitants of the city are referred to as Glaswegians.

A council area is one of the areas defined in Schedule 1 of the Local Government etc. (Scotland) Act 1994 and is under the control of one of the local authorities in Scotland created by that Act.

Edinburgh is the capital city of Scotland and one of its 32 local government council areas. Located in Lothian on the Firth of Forths southern shore, it is Scotlands second most populous city and the seventh most populous in the United Kingdom. The 2014 official population estimates are 464,990 for the city of Edinburgh, 492,680 for the local authority area, and 1,339,380 for the city region as of 2014 (Edinburgh lies at the heart of the proposed Edinburgh and South East Scotland city region). Recognised as the capital of Scotland since at least the 15th century, Edinburgh is home to the Scottish Parliament and the seat of the monarchy in Scotland. The city is also the annual venue of the General Assembly of the Church of Scotland and home to national institutions such as the National Museum of Scotland, the National Library of Scotland and the Scottish National Gallery. It is the largest financial centre in the UK after London.

Carlisle (or from Cumbric: "Caer Luel" ) is a city and the county town of Cumbria. Historically in Cumberland, it is also the administrative centre of the City of Carlisle district in North West England. Carlisle is located at the confluence of the rivers Eden, Caldew and Petteril, south of the Scottish border. It is the largest settlement in the county of Cumbria, and serves as the administrative centre for both Carlisle City Council and Cumbria County Council. At the time of the 2001 census, the

population of Carlisle was 71,773, with 100,734 living in the wider city. Ten years later, at the 2011 census, the citys population had risen to 75,306, with 107,524 in the wider city.

Hamilton is a town in South Lanarkshire, in the central Lowlands of Scotland. It serves as the main administrative centre of the South Lanarkshire council area. It is the fourth-biggest town in Scotland. It sits south-east of Glasgow, south-west of Edinburgh and north of Carlisle, Cumbria. It is situated on the south bank of the River Clyde at its confluence with the Avon Water. Hamilton is the historical county town of Lanarkshire.

South Lanarkshire is one of 32 unitary authorities of Scotland. It borders the south-east of the City of Glasgow and contains some of Greater Glasgows suburbs. It also contains many towns and villages. It also shares borders with Dumfries and Galloway, East Ayrshire, East Renfrewshire, North Lanarkshire, the Scottish Borders and West Lothian. It includes part of the historic county of Lanarkshire.

The Central Lowlands or Midland Valley is a geologically defined area of relatively low-lying land in southern Scotland. It consists of a rift valley between the Highland Boundary Fault to the north and the Southern Uplands Fault to the south. The Central Lowlands are one of the three main geographical sub-divisions of Scotland, the other two being the Highlands and Islands which lie to the north, northwest and the Southern Uplands, which lie south of the associated second fault line.

The River Clyde is a river, that flows into the Firth of Clyde in Scotland. It is the eighth-longest river in the United Kingdom, and the second-longest in Scotland. Flowing through the major city of Glasgow, it was an important river for shipbuilding and trade in the British Empire. In the early medieval Cumbric language it was known as "Clud" or "Clut", and was central to the Kingdom of Strathclyde ("Teyrnas Ystrad Clut").

Scotland (Scots: ) is a country that is part of the United Kingdom and covers the northern third of the island of Great Britain. It shares a border with England to the south, and is otherwise surrounded by the Atlantic Ocean, with the North Sea to the east and the North Channel and Irish Sea to the south-west. In addition to the mainland, the country is made up of more than 790 islands, including the Northern Isles and the Hebrides.

Avon Water, also known locally as the River Avon, is a river in Scotland, and a tributary of the River Clyde.

Lanarkshire, also called the County of Lanark is a historic county in the central Lowlands of Scotland.

### A.4.2 TYPE 2 ERROR: UNANSWERABLE

**Total**   $\frac{28}{100}$

**Query**   `narrative_location the beloved vagabond`

**Candidates**   2014, arctic, atlantic ocean, austin, austria, belgium, brittany, burgundy, cyprus, earth, england, europe, finland, france, frankfurt, germany, hollywood, israel, lithuania, london, luxembourg, lyon, marseille, netherlands, paris, portugal, rhine, swiss alps, victoria, worms

**Answer**   london

**Prediction**   marseille

**Support documents**   The North Sea is a marginal sea of the Atlantic Ocean located between Great Britain, Scandinavia, Germany, the Netherlands, Belgium, and France. An epeiric (or "shelf") sea on the European continental shelf, it connects to the ocean through the English Channel in the south and the Norwegian Sea in the north. It is more than long and wide, with an area of around .

Worms is a city in Rhineland-Palatinate, Germany, situated on the Upper Rhine about south-southwest of Frankfurt-am-Main. It had approximately 85,000 inhabitants .

William George "Will" Barker (18 January 1868 in Cheshunt 6 November 1951 in Wimbledon) was a British film producer, director, cinematographer, and entrepreneur who took film-making in

Britain from a low budget form of novel entertainment to the heights of lavishly-produced epics that were matched only by Hollywood for quality and style .

Ealing is a major suburban district of west London, England and the administrative centre of the London Borough of Ealing. It is one of the major metropolitan centres identified in the London Plan. It was historically a rural village in the county of Middlesex and formed an ancient parish. Improvement in communications with London, culminating with the opening of the railway station in 1838, shifted the local economy to market garden supply and eventually to suburban development.

Paris (French: ) is the capital and most populous city of France. It has an area of and a population in 2013 of 2,229,621 within its administrative limits. The city is both a commune and department, and forms the centre and headquarters of the le-de-France, or Paris Region, which has an area of and a population in 2014 of 12,005,077, comprising 18.2 percent of the population of France.

Bordeaux (Gascon Occitan: ””) is a port city on the Garonne River in the Gironde department in southwestern France.

The euro (sign: ; code: EUR) is the official currency of the eurozone, which consists of 19 of the member states of the European Union: Austria, Belgium, Cyprus, Estonia, Finland, France, Germany, Greece, Ireland, Italy, Latvia, Lithuania, Luxembourg, Malta, the Netherlands, Portugal, Slovakia, Slovenia, and Spain. The currency is also officially used by the institutions of the European Union and four other European countries, as well as unilaterally by two others, and is consequently used daily by some 337 million Europeans . Outside of Europe, a number of overseas territories of EU members also use the euro as their currency.

Lille is a city in northern France, in French Flanders. On the Dele River, near Frances border with Belgium, it is the capital of the Hauts-de-France region and the prefecture of the Nord department.

The Big Pond is a 1930 American Pre-Code romantic comedy film based on a 1928 play of the same name by George Middleton and A.E. Thomas. The film was written by Garrett Fort, Robert Presnell Sr. and Preston Sturges, who provided the dialogue in his first Hollywood assignment, and was directed by Hobart Henley. The film stars Maurice Chevalier and Claudette Colbert, and features George Barbier, Marion Ballou, and Andre Corday, and was released by Paramount Pictures.

Passport to Pimlico is a 1949 British comedy film made by Ealing Studios and starring Stanley Holloway, Margaret Rutherford and Hermione Baddeley. It was directed by Henry Cornelius and written by T. E. B. Clarke. The story concerns the unearthing of treasure and documents that lead to a small part of Pimlico to be declared a legal part of the House of Burgundy, and therefore exempt from the post-war rationing or other bureaucratic restrictions active in Britain at the time.

Lyon or (more archaically) Lyons (or ) is a city in east-central France, in the Auvergne-Rhne-Alpes region, about from Paris and from Marseille. Inhabitants of the city are called ”Lyonnais”.

”Thank Heaven for Little Girls” is a 1957 song written by Alan Jay Lerner and Frederick Loewe and often associated with performer Maurice Chevalier. It opened and closed the 1958 film ”Gigi”. Alfred Drake performed the song in the 1973 Broadway stage production of ”Gigi”, though in the 2015 revival, it was sung as a duet between Victoria Clark and Dee Hoty.

The Lavender Hill Mob is a 1951 comedy film from Ealing Studios, written by T.E.B. Clarke, directed by Charles Crichton, starring Alec Guinness and Stanley Holloway and featuring Sid James and Alfie Bass. The title refers to Lavender Hill, a street in Battersea, a district of South London, in the postcode district SW11, near to Clapham Junction railway station.

Curtis Bernhardt (15 April 1899 22 February 1981) was a German film director born in Worms, Germany, under the name Kurt Bernhardt. He trained as an actor in Germany, and performed on the stage, before starting as a film director in 1926. Other films include ”A Stolen Life” (1946) and ”Sirocco” (1951).

Toulouse is the capital city of the southwestern French department of Haute-Garonne, as well as of the Occitanie region. The city lies on the banks of the River Garonne, from the Mediterranean Sea, from the Atlantic Ocean, and from Paris. It is the fourth-largest city in France with 466,297 inhabitants in January 2014. The Toulouse Metro area is, with 1 312 304 inhabitants as of 2014, Frances 4th metropolitan area after Paris, Lyon and Marseille and ahead of Lille and Bordeaux.

French Guiana (pronounced or ), officially called Guiana, is an overseas department and region of France, located on the north Atlantic coast of South America in the Guyanas. It borders Brazil to the east and south, and Suriname to the west. Its area has a very low population density of only 3 inhabitants per km, with half of its 244,118 inhabitants in 2013 living in the metropolitan area of Cayenne, its capital. By land area, it is the second largest region of France and the largest outermost region within the European Union.

The Mediterranean Sea (pronounced ) is a sea connected to the Atlantic Ocean, surrounded by the Mediterranean Basin and almost completely enclosed by land: on the north by Southern Europe and Anatolia, on the south by North Africa, and on the east by the Levant. The sea is sometimes considered a part of the Atlantic Ocean, although it is usually identified as a separate body of water.

Maurice Auguste Chevalier (September 12, 1888 January 1, 1972) was a French actor, cabaret singer and entertainer. He is perhaps best known for his signature songs, including "Louise", "Mimi", "Valentine", and "Thank Heaven for Little Girls" and for his films, including "The Love Parade" and "The Big Pond". His trademark attire was a boater hat, which he always wore on stage with a tuxedo.

Nice (; Niard , classical norm, or "", nonstandard, ) is the fifth most populous city in France and the capital of the Alpes-Maritimes "dpartement". The urban area of Nice extends beyond the administrative city limits, with a population of about 1 million on an area of . Located in the French Riviera, on the south east coast of France on the Mediterranean Sea, at the foot of the Alps, Nice is the second-largest French city on the Mediterranean coast and the second-largest city in the Provence-Alpes-Cte dAzur region after Marseille. Nice is about 13 kilometres (8 miles) from the principality of Monaco, and its airport is a gateway to the principality as well.

Ealing Studios is a television and film production company and facilities provider at Ealing Green in west London. Will Barker bought the White Lodge on Ealing Green in 1902 as a base for film making, and films have been made on the site ever since. It is the oldest continuously working studio facility for film production in the world, and the current stages were opened for the use of sound in 1931. It is best known for a series of classic films produced in the post-WWII years, including "Kind Hearts and Coronets" (1949), "Passport to Pimlico" (1949), "The Lavender Hill Mob" (1951), and "The Ladykillers" (1955). The BBC owned and filmed at the Studios for forty years from 1955 until 1995. Since 2000, Ealing Studios has resumed releasing films under its own name, including the revived "St Trinians" franchise. In more recent times, films shot here include "The Importance of Being Earnest" (2002) and "Shaun of the Dead" (2004), as well as "The Theory of Everything" (2014), "The Imitation Game" (2014) and "Burnt" (2015). Interior scenes of the British period drama television series "Downton Abbey" are shot in Stage 2 of the studios. The Met Film School London operates on the site.

Kind Hearts and Coronets is a British black comedy film of 1949 starring Dennis Price, Joan Greenwood, Valerie Hobson, and Alec Guinness. Guinness plays eight distinct characters. The plot is loosely based on the novel "Israel Rank: The Autobiography of a Criminal" (1907) by Roy Horniman, with the screenplay written by Robert Hamer and John Dighton and the film directed by Hamer. The title refers to a line in Tennysons poem "Lady Clara Vere de Vere": "Kind hearts are more than coronets, and simple faith than Norman blood."

Europe is a continent that comprises the westernmost part of Eurasia. Europe is bordered by the Arctic Ocean to the north, the Atlantic Ocean to the west, and the Mediterranean Sea to the south. To the east and southeast, Europe is generally considered as separated from Asia by the watershed divides of the Ural and Caucasus Mountains, the Ural River, the Caspian and Black Seas, and the waterways of the Turkish Straits. Yet the non-oceanic borders of Europea concept dating back to classical antiquityare arbitrary. The primarily physiographic term "continent" as applied to Europe also incorporates cultural and political elements whose discontinuities are not always reflected by the continents current overland boundaries.

France, officially the French Republic, is a country with territory in western Europe and several overseas regions and territories. The European, or metropolitan, area of France extends from the Mediterranean Sea to the English Channel and the North Sea, and from the Rhine to the Atlantic Ocean. Overseas France include French Guiana on the South American continent and several island territories in the Atlantic, Pacific and Indian oceans. France spans and had a total population of almost 67 million people as of January 2017. It is a unitary semi-presidential republic with the

capital in Paris, the countrys largest city and main cultural and commercial centre. Other major urban centres include Marseille, Lyon, Lille, Nice, Toulouse and Bordeaux.

The British Broadcasting Corporation (BBC) is a British public service broadcaster. It is headquartered at Broadcasting House in London, is the worlds oldest national broadcasting organisation, and is the largest broadcaster in the world by number of employees, with over 20,950 staff in total, of whom 16,672 are in public sector broadcasting; including part-time, flexible as well as fixed contract staff, the total number is 35,402.

The Rhine (, , ) is a European river that begins in the Swiss canton of Graubnden in the southeastern Swiss Alps, forms part of the Swiss-Austrian, Swiss-Liechtenstein, Swiss-German and then the Franco-German border, then flows through the Rhineland and eventually empties into the North Sea in the Netherlands. The largest city on the river Rhine is Cologne, Germany, with a population of more than 1,050,000 people. It is the second-longest river in Central and Western Europe (after the Danube), at about , with an average discharge of about .

The Beloved Vagabond is a 1936 British musical drama film directed by Curtis Bernhardt and starring Maurice Chevalier , Betty Stockfeld , Margaret Lockwood and Austin Trevor . In nineteenth century France an architect posing as a tramp falls in love with a woman . The film was made at Ealing Studios by the independent producer Ludovico Toeplitz .

The Atlantic Ocean is the second largest of the worlds oceans with a total area of about . It covers approximately 20 percent of the Earths surface and about 29 percent of its water surface area. It separates the "Old World" from the "New World".

Claude Austin Trevor (7 October 1897 22 January 1978) was a Northern Irish actor who had a long career in film and television.

The English Channel ("the Sleeve" [hence ] "Sea of Brittany" "British Sea"), also called simply the Channel, is the body of water that separates southern England from northern France, and joins the southern part of the North Sea to the rest of the Atlantic Ocean.

### A.4.3 TYPE 3 ERROR: MULTIPLE CORRECT ANSWERS

**Total** $\frac{22}{100}$

**Query** `instance_of qilakitsoq`

**Candidates** 1, academic discipline, activity, agriculture, archaeological site, archaeological theory, archaeology, archipelago, architecture, base, bay, branch, century, circle, coast, company, constituent country, continent, culture, director, endangered language, evidence, family, ferry, five, fjord, group, gulf, history, human, humans, hunting, inlet, island, lancaster, language isolate, material, monarchy, municipality, part, peninsula, people, queen, realm, region, republic, science, sea, sign, sound, study, subcontinent, system, territory, theory, time, town, understanding, war, world war, year

**Answer** town

**Prediction** archaeological site

**Support documents** North America is a continent entirely within the Northern Hemisphere and almost all within the Western Hemisphere. It can also be considered a northern subcontinent of the Americas. It is bordered to the north by the Arctic Ocean, to the east by the Atlantic Ocean, to the west and south by the Pacific Ocean, and to the southeast by South America and the Caribbean Sea.

Inuit (pronounced or ; Inuktitut: , "the people") are a group of culturally similar indigenous peoples inhabiting the Arctic regions of Greenland, Canada and Alaska. Inuit is a plural noun; the singular is Inuk. The oral Inuit languages are classified in the Eskimo-Aleut family. Inuit Sign Language is a critically endangered language isolate spoken in Nunavut.

Qilakitsoq is an archaeological site on Nuussuaq Peninsula , on the shore of Uummannaq Fjord in northwestern Greenland . Formally a settlement , it is famous for the discovery of eight mummified bodies in 1972 . Four of the mummies are currently on display in the Greenland National Museum .

Norway (; Norwegian: (Bokml) or (Nynorsk); Sami: "Norgga"), officially the Kingdom of Norway, is a sovereign and unitary monarchy whose territory comprises the western portion of the Scandinavian Peninsula plus the island Jan Mayen and the archipelago of Svalbard. The Antarctic Peter I Island and the sub-Antarctic Bouvet Island are dependent territories and thus not considered part of the Kingdom. Norway also lays claim to a section of Antarctica known as Queen Maud Land. Until 1814, the Kingdom included the Faroe Islands (since 1035), Greenland (1261), and Iceland (1262). It also included Shetland and Orkney until 1468. It also included the following provinces, now in Sweden: Jmtland, Hrjedalen and Bohusln.

The Arctic (or ) is a polar region located at the northernmost part of Earth. The Arctic consists of the Arctic Ocean, adjacent seas, and parts of Alaska (United States), Canada, Finland, Greenland (Denmark), Iceland, Norway, Russia, and Sweden. Land within the Arctic region has seasonally varying snow and ice cover, with predominantly treeless permafrost-containing tundra. Arctic seas contain seasonal sea ice in many places.

Archaeology, or archeology, is the study of human activity through the recovery and analysis of material culture. The archaeological record consists of artifacts, architecture, biofacts or ecofacts, and cultural landscapes. Archaeology can be considered both a social science and a branch of the humanities. In North America, archaeology is considered a sub-field of anthropology, while in Europe archaeology is often viewed as either a discipline in its own right or a sub-field of other disciplines.

An archaeological site is a place (or group of physical sites) in which evidence of past activity is preserved (either prehistoric or historic or contemporary), and which has been, or may be, investigated using the discipline of archaeology and represents a part of the archaeological record. Sites may range from those with few or no remains visible above ground, to buildings and other structures still in use.

Nuussuaq Peninsula (old spelling: "Ngssuaq") is a large (180x48 km) peninsula in western Greenland.

Geologically, a fjord or fiord is a long, narrow inlet with steep sides or cliffs, created by glacial erosion. There are many fjords on the coasts of Alaska, British Columbia, Chile, Greenland, Iceland, the Kerguelen Islands, New Zealand, Norway, Novaya Zemlya, Labrador, Nunavut, Newfoundland, Scotland, and Washington state. Norways coastline is estimated at with fjords, but only when fjords are excluded.

Baffin Bay (Inuktitut: "Saknirutiak Imanga"; ), located between Baffin Island and the southwest coast of Greenland, is a marginal sea of the North Atlantic Ocean. It is connected to the Atlantic via Davis Strait and the Labrador Sea. The narrower Nares Strait connects Baffin Bay with the Arctic Ocean. The bay is not navigable most of the year because of the ice cover and high density of floating ice and icebergs in the open areas. However, a polynya of about , known as the North Water, opens in summer on the north near Smith Sound. Most of the aquatic life of the bay is concentrated near that region. Extent. The International Hydrographic Organization defines the limits of Baffin Bay as follows: History. The area of the bay has been inhabited since . Around 1200, the initial Dorset settlers were replaced by the Thule (the later Inuit) peoples. Recent excavations also suggest that the Norse colonization of the Americas reached the shores of Baffin Bay sometime between the 10th and 14th centuries. The English explorer John Davis was the first recorded European to enter the bay, arriving in 1585. In 1612, a group of English merchants formed the "Company of Merchants of London, Discoverers of the North-West Passage". Their governor Thomas Smythe organized five expeditions to explore the northern coasts of Canada in search of a maritime passage to the Far East. Henry Hudson and Thomas Buttons explored Hudson Bay, William Gibbons Labrador, and Robert Bylot Hudson Strait and the area which became known as Baffins Bay after his pilot William Baffin. Aboard the "Discovery", Baffin charted the area and named Lancaster, Smith, and Jones Sounds after members of his company. By the completion of his 1616 voyage, Baffin held out no hope of an ice-free passage and the area remained unexplored for another two centuries. Over time, his account came to be doubted until it was confirmed by John Rosss 1818 voyage. More advanced scientific

studies followed in 1928, in the 1930s and after World War II by Danish, American and Canadian expeditions.

The archaeological record is the body of physical (not written) evidence about the past. It is one of the core concepts in archaeology, the academic discipline concerned with documenting and interpreting the archaeological record. Archaeological theory is used to interpret the archaeological record for a better understanding of human cultures. The archaeological record can consist of the earliest ancient findings as well as contemporary artifacts. Human activity has had a large impact on the archaeological record. Destructive human processes, such as agriculture and land development, may damage or destroy potential archaeological sites. Other threats to the archaeological record include natural phenomena and scavenging. Archaeology can be a destructive science for the finite resources of the archaeological record are lost to excavation. Therefore archaeologists limit the amount of excavation that they do at each site and keep meticulous records of what is found. The archaeological record is the record of human history, of why civilizations prosper or fail and why cultures change and grow. It is the story of the world that humans have created.

The Danish Realm is a realm comprising Denmark proper, The Faroe Islands and Greenland.

Greenland is an autonomous constituent country within the Danish Realm between the Arctic and Atlantic Oceans, east of the Canadian Arctic Archipelago. Though physiographically a part of the continent of North America, Greenland has been politically and culturally associated with Europe (specifically Norway and Denmark, the colonial powers, as well as the nearby island of Iceland) for more than a millennium. The majority of its residents are Inuit, whose ancestors migrated began migrating from the Canadian mainland in the 13th century, gradually settling across the island.

Uummannaq is a town in the Qaasuitsup municipality, in northwestern Greenland. With 1,282 inhabitants in 2013, it is the eleventh-largest town in Greenland, and is home to the countrys most northerly ferry terminal. Founded in 1763 as maak, the town is a hunting and fishing base, with a canning factory and a marble quarry. In 1932 the Universal Greenland-Filmexpedition with director Arnold Fanck realized the film SOS Eisberg near Uummannaq.

The Republic of Iceland, "Lveldi sland" in Icelandic, is a Nordic island country in the North Atlantic Ocean. It has a population of and an area of , making it the most sparsely populated country in Europe. The capital and largest city is Reykjavk. Reykjavk and the surrounding areas in the southwest of the country are home to over two-thirds of the population. Iceland is volcanically and geologically active. The interior consists of a plateau characterised by sand and lava fields, mountains and glaciers, while many glacial rivers flow to the sea through the lowlands. Iceland is warmed by the Gulf Stream and has a temperate climate, despite a high latitude just outside the Arctic Circle. Its high latitude and marine influence still keeps summers chilly, with most of the archipelago having a tundra climate.

The Canadian Arctic Archipelago, also known as the Arctic Archipelago, is a group of islands north of the Canadian mainland.

Uummannaq Fjord is a large fjord system in the northern part of western Greenland, the largest after Kangertittivaq fjord in eastern Greenland. It has a roughly south-east to west-north-west orientation, emptying into the Baffin Bay in the northwest.

### A.4.4 TYPE 4 ERROR: COMPLEX RELATION TYPES

**Total**   $\frac{8}{100}$

**Query**   `parent_taxon stenotritidae`

**Candidates**   angiosperms, animal, aphid, apocrita, apoidea, area, areas, colletidae, crabronidae, formicidae, honey bee, human, hymenoptera, insects, magnoliophyta, plant, thorax

**Answer**   apoidea

**Prediction**   crabronidae

**Support documents**  A honey bee (or honeybee) is any bee member of the genus Apis, primarily distinguished by the production and storage of honey and the construction of perennial, colonial nests from wax. Currently, only seven species of honey bee are recognized, with a total of 44 subspecies, though historically, from six to eleven species have been recognized. The best known honey bee is the Western honey bee which has been domesticated for honey production and crop pollination. Honey bees represent only a small fraction of the roughly 20,000 known species of bees. Some other types of related bees produce and store honey, including the stingless honey bees, but only members of the genus "Apis" are true honey bees. The study of bees including honey bees is known as melittology.

The superfamily Apoidea is a major group within the Hymenoptera, which includes two traditionally recognized lineages, the "sphecoid" wasps, and the bees. Molecular phylogeny demonstrates that the bees arose from within the Crabronidae, so that grouping is paraphyletic.

Honey is a sugary food substance produced and stored by certain social hymenopteran insects. It is produced from the sugary secretions of plants or insects, such as floral nectar or aphid honeydew, through regurgitation, enzymatic activity, and water evaporation. The variety of honey produced by honey bees (the genus "Apis") is the most well-known, due to its worldwide commercial production and human consumption. Honey gets its sweetness from the monosaccharides fructose and glucose, and has about the same relative sweetness as granulated sugar. It has attractive chemical properties for baking and a distinctive flavor that leads some people to prefer it to sugar and other sweeteners. Most microorganisms do not grow in honey, so sealed honey does not spoil, even after thousands of years. However, honey sometimes contains dormant endospores of the bacterium "Clostridium botulinum", which can be dangerous to babies, as it may result in botulism. People who have a weakened immune system should not eat honey because of the risk of bacterial or fungal infection. Although some evidence indicates honey may be effective in treating diseases and other medical conditions, such as wounds and burns, the overall evidence for its use in therapy is not conclusive. Providing 64 calories in a typical serving of one tablespoon (15 ml) equivalent to 1272 kj per 100 g, honey has no significant nutritional value. Honey is generally safe, but may have various, potential adverse effects or interactions with excessive consumption, existing disease conditions, or drugs. Honey use and production have a long and varied history as an ancient activity, depicted in Valencia, Spain by a cave painting of humans foraging for honey at least 8,000 years ago.

Australia, officially the Commonwealth of Australia, is a country comprising the mainland of the Australian continent, the island of Tasmania and numerous smaller islands. It is the worlds sixth-largest country by total area. The neighbouring countries are Papua New Guinea, Indonesia and East Timor to the north; the Solomon Islands and Vanuatu to the north-east; and New Zealand to the south-east. Australias capital is Canberra, and its largest urban area is Sydney.

Bees are flying insects closely related to wasps and ants, known for their role in pollination and, in the case of the best-known bee species, the European honey bee, for producing honey and beeswax. Bees are a monophyletic lineage within the superfamily Apoidea, presently considered as a clade Anthophila. There are nearly 20,000 known species of bees in seven to nine recognized families, though many are undescribed and the actual number is probably higher. They are found on every continent except Antarctica, in every habitat on the planet that contains insect-pollinated flowering plants.

Solomon Islands is a sovereign country consisting of six major islands and over 900 smaller islands in Oceania lying to the east of Papua New Guinea and northwest of Vanuatu and covering a land area of . The countrys capital, Honiara, is located on the island of Guadalcanal. The country takes its name from the Solomon Islands archipelago, which is a collection of Melanesian islands that also includes the North Solomon Islands (part of Papua New Guinea), but excludes outlying islands, such as Rennell and Bellona, and the Santa Cruz Islands.

The Colletidae are a family of bees, and are often referred to collectively as plasterer bees or polyester bees, due to the method of smoothing the walls of their nest cells with secretions applied with their mouthparts; these secretions dry into a cellophane-like lining. The five subfamilies, 54 genera, and over 2000 species are all evidently solitary, though many nest in aggregations. Two of the subfamilies, Euryglossinae and Hylaeinae, lack the external pollen-carrying apparatus (the scopa) that otherwise characterizes most bees, and instead carry the pollen in their crops. These

groups, and most genera in this family, have liquid or semiliquid pollen masses on which the larvae develop.

Indonesia (or ; Indonesian: ), officially the Republic of Indonesia, is a unitary sovereign state and transcontinental country located mainly in Southeast Asia with some territories in Oceania. Situated between the Indian and Pacific oceans, it is the worlds largest island country, with more than seventeen thousand islands. At , Indonesia is the worlds 14th-largest country in terms of land area and worlds 7th-largest country in terms of combined sea and land area. It has an estimated population of over 260 million people and is the worlds fourth most populous country, the most populous Austronesian nation, as well as the most populous Muslim-majority country. The worlds most populous island of Java contains more than half of the countrys population.

Ants are eusocial insects of the family Formicidae and, along with the related wasps and bees, belong to the order Hymenoptera. Ants evolved from wasp-like ancestors in the Cretaceous period, about 99 million years ago and diversified after the rise of flowering plants. More than 12,500 of an estimated total of 22,000 species have been classified. They are easily identified by their elbowed antennae and the distinctive node-like structure that forms their slender waists.

Tasmania (abbreviated as Tas and known colloquially as "Tassie") is an island state of the Commonwealth of Australia. It is located to the south of the Australian mainland, separated by Bass Strait. The state encompasses the main island of Tasmania, the 26th-largest island in the world, and the surrounding 334 islands. The state has a population of around 518,500, just over forty percent of which resides in the Greater Hobart precinct, which forms the metropolitan area of the state capital and largest city, Hobart.

New Zealand is an island nation in the southwestern Pacific Ocean. The country geographically comprises two main landmassesthat of the North Island, or Te Ika-a-Mui, and the South Island, or Te Waipounamuand numerous smaller islands. New Zealand is situated some east of Australia across the Tasman Sea and roughly south of the Pacific island areas of New Caledonia, Fiji, and Tonga. Because of its remoteness, it was one of the last lands to be settled by humans. During its long period of isolation, New Zealand developed a distinct biodiversity of animal, fungal and plant life. The countrys varied topography and its sharp mountain peaks, such as the Southern Alps, owe much to the tectonic uplift of land and volcanic eruptions. New Zealands capital city is Wellington, while its most populous city is Auckland.

The flowering plants (angiosperms), also known as Angiospermae or Magnoliophyta, are the most diverse group of land plants, with 416 families, approx. 13,164 known genera and a total of c. 295,383 known species. Like gymnosperms, angiosperms are seed-producing plants; they are distinguished from gymnosperms by characteristics including flowers, endosperm within the seeds, and the production of fruits that contain the seeds. Etymologically, angiosperm means a plant that produces seeds within an enclosure, in other words, a fruiting plant. The term "angiosperm" comes from the Greek composite word ("angeion", "case" or "casing", and "sperma", "seed") meaning "enclosed seeds", after the enclosed condition of the seeds.

Pollination is the process by which pollen is transferred to the female reproductive organs of a plant, thereby enabling fertilization to take place. Like all living organisms, seed plants have a single major goal: to pass their genetic information on to the next generation. The reproductive unit is the seed, and pollination is an essential step in the production of seeds in all spermatophytes (seed plants).

Insects (from Latin , a calque of Greek [], "cut into sections") are a class of invertebrates within the arthropod phylum that have a chitinous exoskeleton, a three-part body (head, thorax and abdomen), three pairs of jointed legs, compound eyes and one pair of antennae. They are the most diverse group of animals on the planet, including more than a million described species and representing more than half of all known living organisms. The number of extant species is estimated at between six and ten million, and potentially represent over 90

The Stenotritidae are the smallest of all formally recognized bee families , with only 21 species in two genera , all of them restricted to Australia . Historically , they were generally considered to belong in the family Colletidae , but the stenotritids are presently considered their sister taxon , and deserving of family status . Of prime importance is the stenotritids have unmodified mouthparts , whereas colletids are separated from all other bees by having bilobed glossae . They are large , densely hairy , fast - flying bees , which make simple burrows in the ground and firm , ovoid

provision masses in cells lined with a waterproof secretions . The larvae do not spin cocoons . Fossil brood cells of a stenotritid bee have been found in the Pleistocene of the Eyre Peninsula , South Australia .

A wasp is any insect of the order Hymenoptera and suborder Apocrita that is neither a bee nor an ant. The Apocrita have a common evolutionary ancestor and form a clade; wasps as a group do not form a clade, but are paraphyletic with respect to bees and ants.

