# OpenReview forum: "Coarse-grain Fine-grain Coattention Network for Multi-evidence Question Answering"
_ICLR.cc/2019/Conference_

### Official Review · AnonReviewer2 · 2018-10-31
**I am open to increasing my rating score as long as the authors could address my concerns and confusions below**

**Rating:** 4
**Confidence:** 3

**Review:**

This paper focuses on multi-choice QA and proposes a coarse-to-fine scoring framework. Where the coarse-grained answer scoring model computes the scores with the attention over the whole passages, and the fine-grained one only uses local contexts for each answer option (candidate).

The proposed approach was evaluated on the only dataset of WikiHop, and achieved large improvement over the other methods on the leaderboard. However, I found the paper lack of motivation about the designs of the coarse and fine scoring models. For example, why using self-attention after GRU and co-attention in the two answer scoring models?

Another concern I have is about the novelty. Besides the complicated model designs, the coarse and fine scoring models are both following some common ideas in previous work. And each model could achieve on-par results compared to previous baselines. This makes me feel that the whole approach looks more like model combination of two not-so-novel (and not very well-motivated) models.

Thirdly, the only evaluation on WikiHop brings more problems to the above two points. Since the motivation of the architecture design is not very clear, I am not sure whether the architectures could generalize to other benchmarks. Similar concern for the model combination approach.

Moreover, the proposed approach is a general architecture for multiple-choice datasets requiring multiple evidence. To verify its generalizability, I suggest the authors add further experiments on one dataset from the following ones: either multi-choice QA datasets like ARC and RACE/RACE-open, or other open-domain QA datasets like TriviaQA, by treating the re-ranking of answer predictions as multi-choice QA problems (like the approach in Evidence Aggregation for Open-Domain QA from ICLR2018).

A minor question: why the CFC w/o encoder could still work so well? At least the fine-grained scoring model should heavily rely on encoders. Otherwise, according to Eq (17), the fine-grained model cannot use any contextual information.

---

> ### Author Response · Authors · 2018-11-17
> **Part 1 response to reviewer 3 and additional results on TriviaQA reranking**
>
> We thank you for your detailed feedback! Sorry about breaking this up into 2 parts, it seems we hit the post limit. The results for TriviaQA reranking is in part 2 of this response.
>
> RE: motivation of design of model and novelty
>
> One lesson from prior work on extractive question answering [2, 14] show that the initial encoding of the inputs are crucial to model performance (whereas subsequent decoding matter much less). The coarse-grain and fine-grain modules of the CFC emphasize two complementary approaches to build this initial encoding. The first approach is to read the document with respect to the question, and then find the appropriate answer. The initial encodings produced here are highly relevant to the question, but they do not capture the context in which the answer is mentioned. The second approach is to read the document with respect to the candidate answer, and then consider how relevant this answer is to the question. The initial encodings produced here precisely model the context in which the answer is mentioned, however they do not capture interactions with the question until later. We designed the CFC to incorporate these two complementary approaches into a single network. As a result, our model is able to retain and focus on different aspects of the input (this is shown in our analysis in sections 4.1 and 4.2) and achieve state-of-the-art results on a competitive reading comprehension task.
>
>
> RE: motivation behind the particular usage of self-attention and coattention
>
> We use co-attention to build codependent representations of two inputs. This has been crucial to extractive question answering as well as visual question answering [1-3]. We use self-attention in this work as a means to aggregate information over a variable-length sequence. Prior work have shown its effective across a wide range of NLP tasks [4-8]. An alternative method to information aggregation is pooling, which we compare against in our ablation studies and find to perform worse than self-attention. Our intuition is that self-attention allows for more flexibility in the model and is able to learn task-specific pooling strategies (e.g. emphasize certain regions of the input space over others, as oppose do maximization/minimization/average over features)
>
> The particular design we choose is influenced by what kind of information aggregation we would like to perform. For the coarse-grain module, we would like to first interpret a supporting text with respect to the query, hence we apply coattention over the supporting text and the query. Next, we would like to summarize each variable length supporting text, hence we apply self-attention over each coattention output. Next, we would like to produce a summary of the document collection, which consists of a variable number of supporting texts, hence we apply self-attention again over the set of supporting text summaries. Finally, we query this summary using an answer summary, which we also produce using self-attention. For the fine-grain module, we must first summarize each variable length mention of the answer, hence we apply self-attention. Next, we would like to interpret these mentions with respect to the question, hence we apply coattention. Finally, we would like to summarize this reinterpretation to produce a score, hence we apply self-attention again.

---

> > ### Author Response · Authors · 2018-11-17
> > **Part 2 response to reviewer 3 and additional results on TriviaQA reranking**
> >
> > RE: reranking existing question answering models, as well as other other tasks
> >
> > Thanks for mentioning this! After considering your feedback we ran preliminary experiments in which we applied our model to rerank the outputs produced by (the merge version of) BiDAF++ [13] on TriviaQA [12]. In particular, we rerank the top-50 answers produced by BiDAF++ (let’s refer to the as the “N-best list”). We find that on the subset of the dev set in which the ground-truth answer is present in the N-best list (which is 86.8% of the data), reranking using our method improves exact match accuracy (EM) from 59.8% to 63.2% and F1 from 64.5% to 67.8%. On the subset in which the ground-truth answer is not present in the N-best list (which is 13.2% of the data), reranking using our method improves EM from 17.5% to 18.4% and F1 from 22.2% to 23.2%. On the entire dev set, reranking using our method improves EM from 54.0% to 57.1% and F1 from 58.7% to 61.7%. Hence, we think our method can be used to improve existing span-extractive models as a reranker (e.g. similar to [9-10]). We will add this result to the paper.
> >
> > The setup in the ARC/RACE/RACE-open are such that
> > - The evidence is a collection of short sentences (a key challenge is IR)
> > - The answers are propositions whose truth values are determined by evidence in the sentences
> >
> > In contrast, Qangaroo is such that
> > - The evidence is a collection of paragraphs obtained from relation-guided knowledge graph traversal
> > - The answers are plausible entities for the relation in interest
> >
> > While we think both of these settings are interesting, our model is designed for the latter and unsuitable for the former since the the answers are not entities, so the fine-grain module does not have entity mentions to attach to. One can hypothetically extract entities from non-entity answers, however for ARC/RACE/RACE-open, multiple answer choices refer to the same entity (e.g. he did A vs. he did B). That is, the mentions are shared between answers and do not distinguish between answers. Consequently the output of the fine-grain module for answers that share entities would be the same.
> >
> >
> > RE: question on encoder-less ablation
> >
> > The fine-grain module still has a bidirectional GRU inside the coattention (Please see eq. 18 and eq. 7)
> >
> > [1] https://arxiv.org/abs/1606.00061
> > [2] https://arxiv.org/abs/1611.01604
> > [3] https://arxiv.org/abs/1711.00106
> > [4] https://arxiv.org/abs/1801.10296
> > [5] https://arxiv.org/abs/1805.01052
> > [6] https://arxiv.org/abs/1707.07045
> > [7] https://arxiv.org/abs/1706.03762
> > [8] https://arxiv.org/abs/1805.09655
> > [9] https://arxiv.org/abs/1808.05759
> > [10] https://arxiv.org/abs/1711.05116
> > [11] https://arxiv.org/abs/1805.08092
> > [12] https://arxiv.org/abs/1705.03551
> > [13] https://arxiv.org/abs/1710.10723
> > [14] https://arxiv.org/abs/1611.01603

---

> > ### Comment · AnonReviewer2 · 2018-11-26
> > **This does not address my concerns**
> >
> > In my review, I wrote "since the motivation of the architecture design is not very clear, I am not sure whether the architectures could generalize to other benchmarks". I absolutely understand the thoughts behind your model design. And I am looking for evidence that this specific design really generalizes to other QA datasets. I also raised similar concern on the coarse-fine model as "similar concern for the model combination approach".
> >
> > However, in the newly added experiments on TriviaQA, the proposed model architectures were not compared with the other models. And I did not find a baseline multi-evidence or re-ranking technique compared in the table (please correct me if I am wrong). This makes me a little confused on the experimental design on TriviaQA. In my mind ideally the new experiments should show (1) the proposed coarse-model is at least comparable to the other multi-evidence and re-ranking techniques (which seems not by looking at the leaderboard of TriviaQA-Wiki, also note that here only dev results are reported); (2) the coarse+fine re-ranking should be significantly better compared to using only coarse model. But from the current version, I did not find evidence supporting my above two hypotheses.
> >
> > One quick question: could you please let me know what numbers on TriviaQA-Wiki I should compare your numbers with?
> >
> > Please let me know if you have further results supporting the above hypotheses, or if you think the above hypotheses are not important for the experimental design.

---

> > > ### Author Response · Authors · 2018-11-27
> > > **We hope that our experiments on TriviaQA shows that our proposed method generalizes to other tasks in addition to obtaining state-of-the-art results on Qangaroo.**
> > >
> > > We agree with you that this paper will be strengthened by showing that our method generalizes to other tasks. We hope that our experiments on TriviaQA shows that our proposed method generalizes to other tasks in addition to obtaining state-of-the-art results on Qangaroo.
> > >
> > > It is not our intention to claim state-of-the-art performance on reranking extractive question answering systems. The leading methods on the TriviaQA Wiki leaderboard are unpublished or do not have open-source implementation. Rerankers are also evaluated on different setups (none on TriviaQA Web or Wiki leaderboard). It would take significant amount of effort to consolidate the differences, reproduce state-of-the-art rerankers, and compare results. Instead, we showed that our technique, without modifications, leads to significant improvement in reranking a competitive extractive model on TriviaQA as you suggested, which shows our method generalizes beyond Qangaroo. We also ran the second experiment you suggested. Our results show that the coarse+fine model outperforms the coarse-only model on reranking as well. The former obtains 57.1 EM and 61.7 F1 whereas the latter obtains 54.2 EM and 58.9 F1.

---

> > > > ### Comment · AnonReviewer2 · 2018-11-27
> > > > **Thanks for the new result of coarse-only model**
> > > >
> > > > Thanks for posting the new result. That is very helpful. It makes sense that the coarse-only model does not help on this task but the fine-grain model is much more useful. I will discuss with the other reviewers asap to make my final decision.

---

> > > > > ### Author Response · Authors · 2018-11-27
> > > > > **Thank you**
> > > > >
> > > > > Thank you for your prompt reviews and responses!

---

### Official Review · AnonReviewer3 · 2018-11-01
**Good work; could be more interesting to see application in other related tasks**

**Rating:** 7
**Confidence:** 5

**Review:**

This paper proposes an interesting coarse-grain fine-grain coattention network architecture to address multi-evidence question answering and achieves the new state-of-the-art results on the Qangaroo WikiHop dataset.  The main idea is to divide the task across the coarse-grain and fine-grain modules in a complimentary manner such that the coarse-grain module learns from efficient modeling of support documents and the query whereas the fine-grain module learns from associations of candidate mentions in the support documents with the query.

The major strength of the model is observed with learning effective representations of larger numbers of long support documents and the state-of-the-art results are achieved without the use of pretrained contextualized embeddings. The main novelty lies in how the coattention and self-attention strategies are combined hierarchically to learn relevant representations in a complimentary fashion (rather than serial). Overall, the paper is very well-written and presents solid results with meaningful ablation study, quantitative and qualitative analyses. I have a few comments/suggestions:

- It would be interesting to see how the inclusion of pretrained contextualized embeddings such as ELMo, ULMFit, BERT would help the current model.

- "This is likely because coreference resolution captures intra-document and inter-document dependencies more accurately than hierarchical attention." --> Please clarify why this is the case.

- "We hypothesize that ways to reduce this type of error include using more robust pretrained contextual encoders (McCann et al., 2017; Peters et al., 2018) and coreference resolution." --> I agree; also, it would be worth considering some commonsense knowledge to alleviate this issue, because the fact that Scotland is a part of UK and has a border with England should be learned. Here is a relevant work: "Knowledgeable Reader: Enhancing Cloze-Style Reading Comprehension with External Commonsense Knowledge" by Mihaylov and Frank, 2018.

- "The second type (28% of errors) results from questions that are not answerable. For example, the support documents do not provide the narrative location of the play “The Beloved Vagabond” for the query narrative location the beloved vagabond." --> It would be great if you could release the set of unanswerable questions for the community.

- Please include the memory network-based QA works in the related work section because they involve some forms of reasoning. Also, I would suggest to cover the query-focused multi-document summarization area in the related work section because they also require evidence synthesis from multiple documents to address a query. It would be very interesting if authors can apply their model for the query-focused multi-document summarization task as well, as this would further validate the effectiveness of the proposed architecture for reasoning across multiple documents.

---

> ### Author Response · Authors · 2018-11-17
> **Response to reviewer 2 comments**
>
> We thank you for your feedback, and are glad that you found the results to be useful and manuscript to be well-written and meaningfully ablated and analyzed.
>
> RE: incorporation of world knowledge and common-sense knowledge
>
> Thank you for the suggestion!
>
> Zellers et al. [1] find that contextual embeddings capture a surprising amount of common sense knowledge (please see the EMNLP 2018 presentation on new SWAG results [2]). Perhaps this accounts for the consistent gains brought about by CoVe [3], ELMo [4], and BERT [5] on a wide variety of NLP tasks.
>
> Unfortunately we could not do these experiments in time due to computational constraints. In particular, on-the-fly evaluation of contextualized embeddings required too much memory given the large documents lengths in Qangaroo. We plan to look into practical ways of using these embeddings in future work.
>
>
> RE: coref captures dependencies more accurately than hierarchical attention
>
> We mean that the type of coref we use (e.g. entity matching) introduces very precise direct links, whereas links implicitly captured by soft hierarchical attention is gated and hence noisy. We will clarify this in the writing.
>
>
> RE: list of unanswerable questions
>
> We agree and will release a list of unanswerable questions found during our manual inspection of a subset of the dataset.
>
>
> RE: reference to related works in memory network-based QA and query-focused multi-document summarization
>
> Thank you bringing this to our attention. We will include these in our related works section. We’ll also explore the applicability of our method to query-focused multi-document summarization in the future.
>
> After receiving our reviews, we experimented with our model on another task of reranking extractive question answering. We ran preliminary experiments in which we applied our model to rerank the outputs produced by (the merge version of) BiDAF++ [9] on TriviaQA [8]. In particular, we rerank the top-50 answers produced by BiDAF++ (let’s refer to the as the “N-best list”). We find that on the subset of the dev set in which the ground-truth answer is present in the N-best list (which is 86.8% of the data), reranking using our method improves exact match accuracy (EM) from 59.8% to 63.2% and F1 from 64.5% to 67.8%. On the subset in which the ground-truth answer is not present in the N-best list (which is 13.2% of the data), reranking using our method improves EM from 17.5% to 18.4% and F1 from 22.2% to 23.2%. On the entire dev set, reranking using our method improves EM from 54.0% to 57.1% and F1 from 58.7% to 61.7%. Hence, we think our method can be used to improve existing span-extractive models as a reranker (e.g. similar to [6-7]). We will add this result to the paper.
>
> [1] https://arxiv.org/abs/1808.05326
> [2] https://drive.google.com/file/d/1nRJDlDNVsbBf75tmYIwZj48HM9l4kIxA/view?usp=sharing
> [3] https://arxiv.org/abs/1708.00107
> [4] https://arxiv.org/abs/1802.05365
> [5] https://arxiv.org/abs/1810.04805
> [6] https://arxiv.org/abs/1808.05759
> [7] https://arxiv.org/abs/1711.05116
> [8] https://arxiv.org/abs/1705.03551
> [9] https://arxiv.org/abs/1710.10723

---

> > ### Comment · AnonReviewer3 · 2018-11-26
> > **thanks**
> >
> > Thank you authors for your response and revising the paper based on my comments. Good work!

---

### Official Review · AnonReviewer1 · 2018-11-02
**Method that is well adapted to the task to be solved; clear and well written**

**Rating:** 7
**Confidence:** 4

**Review:**

This paper proposes a method for multi-hop QA based on two separate modules, which are called coarse-grained and fine-grained modules. The coarse-grained module reads all of the supporting documents for QA, whereas the fine-grained one reads the local context surrounding each candidate entity's mentions. Both modules are used to predict the score of a candidate entity being the answer, with the final result being the sum of the two scores.

This is a fine paper and achieves a new state of the art on the Qangaroo multi-hop QA dataset. The paper is clearly written, presents the models intuitively, while not foregoing technical detail should that be interesting to a reader. I appreciated the ablation results, as well as the qualitative analyses. The overall idea of encoding different levels of context is an important one, and I am glad that this paper shows that this approach works for a complex QA task.

There are two downsides to the paper. The first is that I am not sure it is really accurate to call the coarse-grained model as such, as it still seems to require passing every word in the supporting documents to an encoder. It seems to be more aimed at capturing global information from the supporting documents, rather than to make a quick, high-level pass at inference. The second weakness is that the coarse- and fine-grained modules barely interact at all, as their prediction scores are simply summed at the output layer. It is nice that even such a simple method of interaction already works so well, but I would have expected some exploration or comment on how more interactions could be enabled.

---

> ### Author Response · Authors · 2018-11-17
> **Response to reviewer 1 comments**
>
> We thank you for your feedback, and are glad that you found the results to be useful and manuscript to be clearly written and meaningfully ablated and analyzed.
>
>
> RE: the name “coarse-grain”
>
> We chose the word “coarse” for the act of summarizing the entire document collection without observing the question. That is, the resulting summary must compact the documents into a high-level representation without observing the candidate answer (hence “coarse”). In contrast, the fine-grain module summarizes with respect to the candidate answer, hence the summary is more precise (hence “fine”).
>
> RE: lack of interactions between fine-grain and coarse-grain modules
>
> While the modules do not directly interact, the encoders and the embeddings are shared (please see Figure 1). We do agree that finding avenues for interactions is an interesting area for future work. For example, one can extract mention representations from the coattention representations of the coarse-grain module instead of the support encoder.

---

### Author Response · Authors · 2018-11-18
**Summary of updates**

Dear reviewers,

We thank you sincerely for your feedback! We have updated the draft with the information below. A more detailed update can be found in our individual responses to the reviews.

1. We added additional experiments that demonstrate the effectiveness of our method on TriviaQA. In particular, reranking the BiDAF++ span extraction model with our model provides a gain of 3.1% EM and 3.0% F1 on the Wiki dev set.
2. We added the IDs of examples used in the error analysis to the Appendix.
3. We added references to related work in memory networks and query-based multi-document summarization.

---

### Meta-Review · Area_Chair1 · 2018-12-13

**Confidence:** 4
**Recommendation:** Accept (Poster)

**Metareview:**

The paper presents a method for coarse and fine inference for question answering.  It originally measured performance only on WikiHop and then later added experiments on TriviaQA.  The results are good.

One of the concerns regarding the paper was the novelty of the work, and lack of enough experiments.  However, the addition of TriviaQA results allays some of that concern.  I'd suggest citing the paper by Swayamdipta et al from last year that attempted coarse to fine inference for TriviaQA:

Multi-Mention Learning for Reading Comprehension with Neural Cascades.
Swabha Swayamdipta, Ankur P. Parikh and Tom Kwiatkowski.
Proceedings of ICLR 2018.

Overall, there is relative consensus that the paper is good with a new method and some strong results.